# SemHiTok: A Unified Image Tokenizer via Semantic-Guided Hierarchical Codebook for Multimodal Understanding and Generation

**Zisheng Chen**[1,*]**, Chunwei Wang**[2]**, Runhui Huang**[3]**, Hongbin Xu**[4]**, Xiuwei Chen**[1]**,
Jun Zhou**[1]**, Jianhua Han**[2]**, Hang Xu**[2]**, Xiaodan Liang**[1,†]

[1] Sun Yat-sen University    [2] Huawei Noah's Ark Lab
[3] University of Hong Kong    [4] South China University of Technology

[*]Work done as an intern at Huawei Noah's Ark Lab.
[†]Corresponding Author

## Abstract

In this paper, we introduce **SemHiTok**, a unified image **Tok**enizer via **Sem**antic-Guided **Hi**erarchical codebook that provides consistent discrete representations for multimodal understanding and generation. Recently, unified image tokenizers have sparked exploration within the research community, which is designed to capture high-level semantic features for understanding and retaining low-level pixel features for generation. Previous works attempt to train a unified image tokenizer by combining loss for semantic distillation and pixel reconstruction. However, due to the differing levels of features prioritized by multimodal understanding and generation, joint training methods face significant challenges in achieving a good trade-off. SemHiTok addresses this challenge through a novel semantic-guided hierarchical codebook, which builds pixel sub-codebooks on a pretrained semantic codebook. This design decouples the semantic and pixel in terms of structure and training strategy, enabling the tokenizer to capture pixel features while retaining its ability to comprehend high-level semantic information. Our experiments demonstrate that SemHiTok achieves leading performance in image reconstruction and multimodal understanding under the LLaVA-v1.5 setting. Further, we develop a unified MLLM with SemHiTok, which exhibits superior performance across multimodal understanding and generation tasks. Extensive experiments confirm our analysis, showing that our unified image tokenizer architecture achieves a better trade-off.

## 1 Introduction

In recent years, autoregressive models have achieved great success in natural language processing and have been extended to the multimodal understanding domain, demonstrating immense potential. This triggers researchers' interest in unified multimodal understanding and generation by employing a single autoregressive framework. To achieve a unified multimodal large model, the key challenge is designing a tokenizer suitable for both multimodal generation and understanding tasks.

However, there is a vast gap in the visual information required for these two tasks. For instance, models from the CLIP (Radford et al., 2021a; Sun et al., 2023; Zhai et al., 2023) family, commonly used in multimodal understanding tasks, tend to lose visual pixel information. On the contrary, the VQGAN (Yu et al., 2021a; Zhu et al., 2024a) family models, often used in autoregressive generation tasks, lack the ability to extract semantic features for multimodal understanding tasks. This leads to poor performance when a single tokenizer is applied to a unified MLLM (Wu et al., 2024b; Jin et al., 2023; Li et al., 2024b). In light of the aforementioned issues, some recent work has attempted to incorporate a semantic learning branch into the original VQGAN training pipeline, aiming to obtain a unified tokenizer via joint optimization. VILA-U (Wu et al., 2024c) employs a straightforward combination of semantic alignment loss and pixel reconstruction loss, which allows the model to

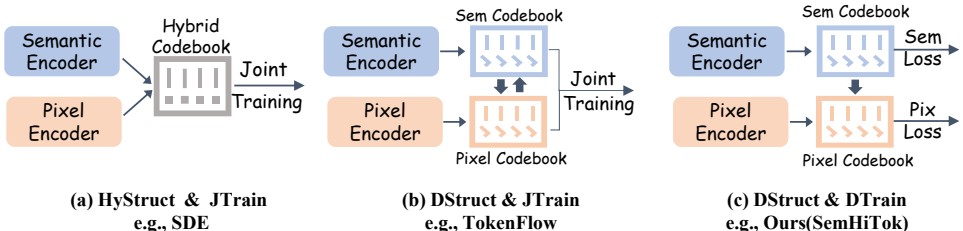

Figure 1: Illustration of other tokenizers and SemHiTok. **HyStruct**: Using a single model to extract information at different levels; **DStruct**: Using different models to extract information at various levels; **JTrain**: Using a joint optimization training strategy; **DTrain**: Adopt a phased optimization training strategy.

capture both low-level and high-level information. Nevertheless, the reliance on a hybrid structure and joint optimization often drives the tokenizer toward a suboptimal solution. Furthermore, some recent works (Xie et al., 2025; Qu et al., 2024) have built upon it with further improvements. As shown in Fig.1(a), while SDE (Xie et al., 2025) further decouples the encoders, the remaining hybrid codebook continues to impede the model's optimization. TokenFlow (Qu et al., 2024) uses shared mapping to decouple the semantic branch and pixel branch while maintaining the consistency of the codebook index, but joint training still affects the final performance.

A straightforward approach is to use CLIP and VQGAN to extract semantic and pixel information, respectively, and the concatenation of these two token sequences is then used as a unified representation. Janus (Wu et al., 2024a) introduces a dual-encoder method that separates encoders for understanding and generation tasks to address this conflict, but this increases the complexity of handling mixed tasks and does not fundamentally resolve the feature conflict challenge. However, this leads to a doubling of the token sequence count or multiplicative expansion in vocabulary size, depending on whether the concatenation is applied along the length or dimension. These limitations underscore a fundamental challenge in the field: *How to balance semantic-level and pixel-level information effectively, without compromising the ease of integration into MLLM frameworks?*

To address this challenge, we propose **SemHiTok**, a unified image tokenizer that provides consistent feature representations for multimodal understanding and generation tasks through a unique hierarchical codebook design. Inspired by the observation that image patches with the same semantic code tend to have similar pixel features, we introduce a novelty hierarchical codebook which uses a sub-codebook to model the pixel-level space associated with each semantic code, named Semantic-Guided Hierarchical Codebook(SGHC). Unlike existing approaches, SemHiTok supports a stage-wise training paradigm where each stage exclusively optimizes specific hierarchy level features, allowing us to achieve a better trade-off between semantic and pixel feature extraction. In addition, SemHiTok can be seamlessly integrated into existing MLLMs following the next-token paradigm through a simple codebook flattening operation.

Our contributions can be summarized as follows: **(1)**: A novel unified tokenizer that achieves a trade-off between semantic and pixel information, demonstrating outstanding performance in both image reconstruction and multimodal understanding tasks. **(2)**: We develop a unified MLLM architecture that demonstrates superior performance across both multimodal understanding and generation tasks, validating its versatility. **(3)**: Our approach further pushes the performance boundary of unified discrete MLLMs, enabling improved scalability and representation capacity within next-token prediction frameworks.

## 2 METHOD

The main objective of **SemHiTok** is to establish a simple and unified image tokenizer for multimodal understanding and generation. In this model, the image is transformed into discrete tokens that contain semantic information and pixel information. We begin with a semantic codebook training recipe that reconstructs semantic features extracted from a language image pre-training model (Zhai et al., 2023; Wei et al., 2022), and point out the semantic codebook's poor texture feature representation in

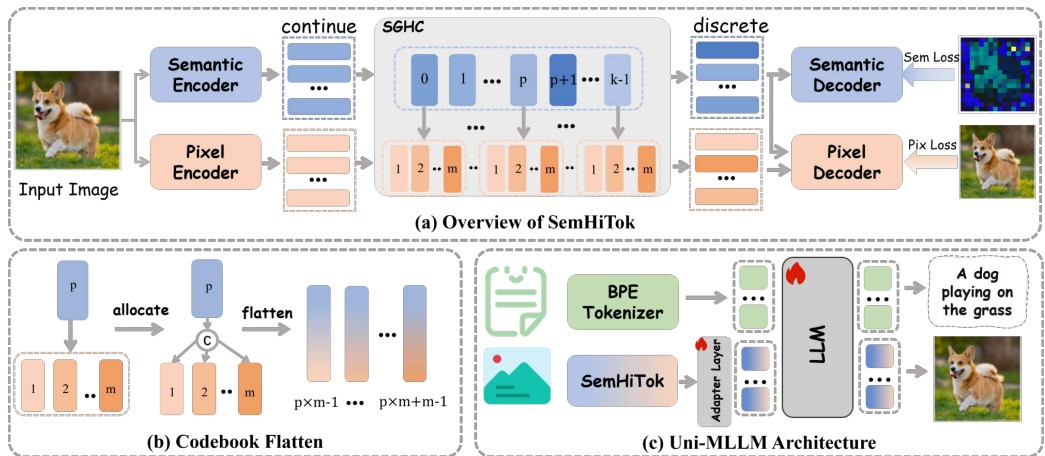

Figure 2: (a) SemHiTok is structurally composed of two branches: **semantic branch** and **pixel branch**. The **semantic branch** is trained following the VQKD (Wei et al., 2022), where the semantic codebook is learned through semantic loss. We propose a semantic-guided hierarchical codebook(SGCH) composed of multiple pixel sub-codebooks, in which each pixel sub-codebook is in a one-to-one correspondence with a semantic code. The selection of pixel sub-codebook is indexed by the semantic code from semantic quantization. To enable a unified discrete representation, we concatenate the quantized semantic and pixel features along the channel dimension and feed the result into the pixel decoder for reconstruction. (b) Each semantic code is allocated to the corresponding pixel sub-codebook, and their features are concatenated along the dimension. (c) An illustration of the unified MLLM framework.

section 2.1. In section 2.2, we conduct a preliminary discussion and observation. Building on this observation, we introduce Semantic-Guided Hierarchical Codebook(SGHC), incorporating texture information while perfectly inheriting the semantic information of the semantic codebook, to enable pixel reconstruction enablement. Finally, we introduce the application of SemHiTok on unified MLLM in section 2.3. The overview framework is shown in Fig.2.

## 2.1 SEMANTIC CODEBOOK TRAINING

For multimodal understanding, using a text-aligned visual encoder (Zhai et al., 2023; Radford et al., 2021a; Sun et al., 2023; Wang et al., 2024b) as an image tokenizer can accelerate convergence and improve performance. However, these text-aligned visual encoders typically output continuous semantic features. In this work, to achieve a unified visual tokenizer, we first train a semantic codebook to quantize the continuous semantic feature following VQKD (Wei et al., 2022).

Given an image $X^{H \times W \times 3}$, the semantic encoder $\mathcal{E}_{\text{sem}}$ extract continuous semantic features:

$$Z_{sem} = \mathcal{E}_{\text{sem}}(X) \in \mathbb{R}^{h \times w \times d_{sem}} \tag{1}$$

Where $\mathcal{E}_{\text{sem}}$ is a frozen text-aligned image encoder, *e.g.*, CLIP (Radford et al., 2021a) or SigLIP (Zhai et al., 2023). Then $Z_{sem}$ are transformed into discrete feature space $\mathcal{C}_{sem} = \{c_1, c_2, ..., c_K\} \in \mathbb{R}^{K \times d_{sem}}$ through quantization function $\mathcal{Q}_{sem}(*)$. The quantization process $\mathcal{Q}_{sem}(*)$ is as follows:

$$Z_{q_{sem}}, I_{q_{sem}} = \underset{k \in \{1, ..., K\}}{\arg\min} \|Z_{sem} - \mathcal{C}_{sem}[k]\| \tag{2}$$

Where $I_{q_{sem}} \in [\mathcal{C}_{sem}]^{h \times w}$ is quantized index, $Z_{q_{sem}}$ is discrete feature indexed from $\mathcal{C}_{sem}$. Finally, semantic decoder $\mathcal{D}_{sem}$ maps $Z_{q_{sem}}$ to raw semantic feature space $\hat{Z}_{sem}$. The $\mathcal{D}_{sem}$ are end-to-end trainable by minimizing semantic distill loss:

$$L_{sem} = 1 - \cos(Z_{q_{sem}}, \hat{Z}_{sem}) + |Z_{q_{sem}} - \hat{Z}_{sem}| \tag{3}$$

For $\mathcal{C}_{sem}$, we adopt EMA (Hunter, 1986) VQ as the semantic codebook. Unlike traditional quantization methods, the EMA VQ is not updated via gradient descent, but instead through an Exponential

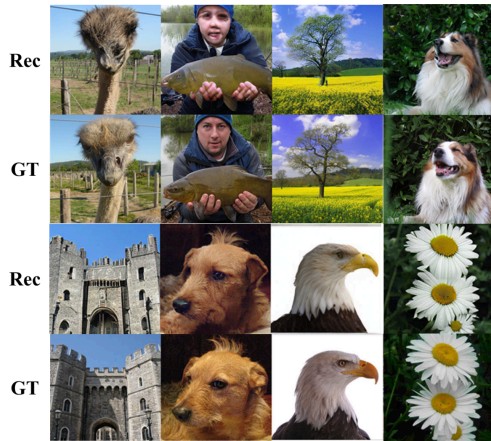

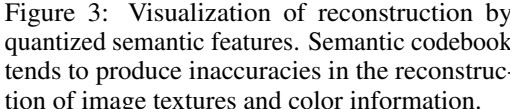

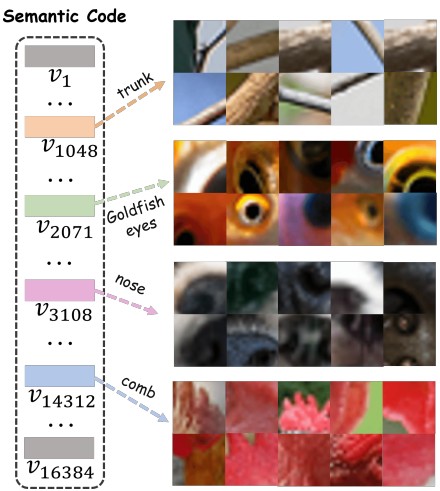

Figure 3: Visualization of reconstruction by quantized semantic features. Semantic codebook tends to produce inaccuracies in the reconstruction of image textures and color information.

Figure 4: Visualization of semantic code. Each code corresponds to a set of image patches that share similar pixel-level features.

Moving Average (EMA) algorithm:

$$\mathbf{c}_k^{(t)} = m \cdot \mathbf{c}_k^{(t-1)} + (1-m) \cdot \frac{1}{N_k} \sum_{i=1}^{N_k} \mathbf{z}_i, \tag{4}$$

where $\mathbf{c}_k^t$ denotes the $k$-th codebook vector at update step $t$, $m$ is the momentum term, and the update is based on the average of all input vectors $\mathbf{z}_i$ assigned to code $\mathbf{k}$ in the current batch. However, we further conduct an experiment that reconstructs the original pixel from the quantized semantic features extracted by $\mathcal{C}_{sem}$. The reconstructed images exhibited noticeable blurriness and a significant loss of high-frequency details, as shown in Fig. 3. This indicates that the semantic codebook lacks pixel information.

## 2.2 PIXEL RECONSTRUCTION ENABLEMENT

**Discussion.** In section 2.1, we demonstrate that semantic code lacks the ability to model pixel information. In order to pixel reconstruction enablement and avoid a reduction of understand ability, a straightforward approach is to add an extra VQGAN (Yu et al., 2021a) model. Semantic codebook extracts discrete semantic tokens for multimodal understanding, and VQGAN extracts discrete texture tokens for generation. The two token sets are concatenated—either dimensionally or sequentially, and passed to the LLM. However, the resulting token inflation or oversized codebook introduces a significant computational burden, limiting its feasibility for MLLMs.

Furthermore, we present the visualization results of the semantic code, as shown in Fig.4. It can be observed that image patches corresponding to the same code exhibit similar pixel features. For example, the code $v_{14312}$ is more likely to be assigned to the rooster comb element in the image. At the same time, the image patches corresponding to these combs exhibit similar pixel features, such as color, patterns, and shapes. Based on this observation, we propose Semantic-Guided Hierarchical Codebook to model the pixel feature space corresponding to each semantic code using a sub-codebook.

**Semantic-Guided Hierarchical Codebook (SGHC).** The SGHC consists of a pretrained semantic codebook and several sub-codebooks, where each sub-codebook corresponds to a semantic code of the semantic codebook, as shown in figure 2 (a). Specially, given the pre-trained semantic codebook $\mathcal{C}_{sem} = \{c_1, c_2, ..., c_K\} \in \mathbb{R}^{K \times d_{sem}}$, the pixel codebook $\mathcal{C}_{pix} = \{\mathcal{C}_{pix}^1, \mathcal{C}_{pix}^2, ..., \mathcal{C}_{pix}^K\} \in \mathbb{R}^{K \times m \times d_{pix}}$, where $\mathcal{C}_{pix}^k \in \mathbb{R}^{m \times d_{pix}}$ is $k_{th}$ semantic code's pixel sub-codebook, $m$ is sub-codebook size. At first, the semantic codebook quantizes $X$ to a discrete semantic token $Z_{sem}$ and a semantic codebook index $I_{sem}$. In parallel, pixel encoder $\mathcal{E}_{pix}$ extract continuous pixel features $Z_{pix} =$

Table 1: Comparison of reconstruction quality on the ImageNet-50k validation set. $^{\ddagger}$: quantizer uses residual quantization (RQ), where the total Code Shape is multiplied by RQ depth. $^{\dagger}$: quantizer uses multiple codebooks and product quantization.

| Method | Res | Code Shape | Codebook Size | rFID↓ |
|---|---|---|---|---|
| *Only Reconstruction* | | | | |
| LlamaGen (Sun et al., 2024a) | 256 | $16 \times 16$ | 16,384 | 2.19 |
| RQVAE (Lee et al., 2022) | 256 | $16 \times 16 \times 4^{\ddagger}$ | 16,384 | 3.20 |
| VQGAN-LC (Zhu et al., 2024a) | 256 | $16 \times 16$ | 100,000 | 2.62 |
| IBQ (Shi et al., 2024) | 256 | $16 \times 16$ | 16,384 | 1.37 |
| IBQ (Shi et al., 2024) | 256 | $16 \times 16$ | 262,144 | 1.00 |
| FQGAN (Bai et al., 2024) | 256 | $16 \times 16 \times 2$ | $16,384 \times 2^{\dagger}$ | 0.94 |
| *Unified* | | | | |
| VILA-U (Wu et al., 2024c) | 256 | $16x16x4^{\ddagger}$ | 16,384 | 1.80 |
| SDE(MUSE-VL) (Xie et al., 2025) | 256 | $16 \times 16$ | 32,768 | 2.26 |
| TokenFlow (Qu et al., 2024) | 256 | 680 | 32,768 | 1.37 |
| TokLIP (Lin et al., 2025) | 256 | $16 \times 16$ | 16,384 | 2.19 |
| QLIP-B (Lin et al., 2025) | 256 | $16 \times 16$ | $2^{28}$ | 3.21 |
| UniTok (Ma et al., 2025) | 256 | $16 \times 16$ | $16,384 \times 4^{\dagger}$ | 0.39 |
| **SemHiTok(ours)** | 256 | $16 \times 16$ | 196,608 | 1.16 |
| **SemHiTok(ours)** | 384 | $27 \times 27$ | 196,608 | 0.66 |

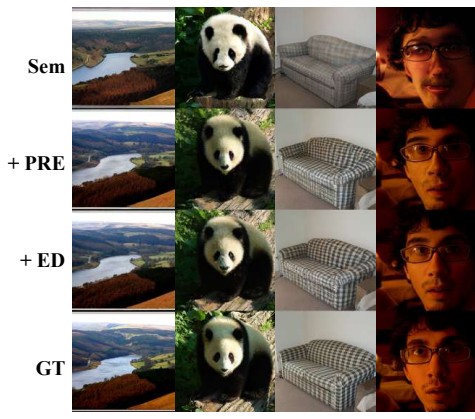

Figure 5: Visualized reconstruction results from the ablation of key modules. PRE brings about a significant improvement in reconstruction quality. Moreover, the Enhance Decoder(ED) further improves reconstruction on hard samples.

$\mathcal{E}_{pix}(X)$. For the quantization process of the pixel codebook, the corresponding pixel sub-codebook is selected based on the quantization result of the semantic codebook. For instance, given image patch $i$, its semantic quantization codebook index $k$ and continuous pixel feature $Z_{pix}^i$, SGHC selects pixel sub-codebook $\mathcal{C}_{pix}^k$ to quantize $Z_{pix}^i$. The process is as follows:

$$Z_{q_{pix}}^i, I_{q_{pix}}^i = \arg\min_{j \in \{1,...,m\}} \|Z_{pix}^i - \mathcal{C}_{pix}^k[j]\| \tag{5}$$

where $j$ is the selected sub-codebook internal index. Finally, the semantic quantized tokens and pixel quantized tokens are concatenated to $Z_q = concate_{dim}(Z_{q_{sem}}, Z_{q_{pix}})$ as the input of pixel decoder $\mathcal{D}_{pix}$ to reconstruct the raw pixel image:

$$\hat{X} = \mathcal{D}_{pix}(Z_q) \tag{6}$$

where $\hat{X}$ is reconstructed pixel image. The $\mathcal{E}_{pix}$, $C_{pix}$ and $\mathcal{D}_{pix}$ are end-to-end trainable by minimizing reconstruction loss $L_{img} = \ell_1(\hat{X}, X)$, codebook loss $Lc$, perceptual loss $L_{per}$ and represents adversarial loss $L_{gan}$ (Yu et al., 2021a). The reconstruction loss is formulated as:

$$L_{rec} = L_{img} + \lambda_1 Lc + \lambda_2 L_{per} + \lambda_3 L_{gan} \tag{7}$$

where $\lambda_1$, $\lambda_2$ and $\lambda_3$ are loss weight of each item.

Our SGHC can be regarded as the refinement of a semantic discrete space to enable pixel reconstruction. We place a specific emphasis on two key advantages of SGHC: (1) ***Non-Conflicting Extension:*** Our method leverages a pre-trained semantic codebook as a foundation, with pixel reconstruction losses exclusively employed to optimize pixel branch modules during the PRE. This strategic approach effectively circumvents the suboptimal solutions that arise from joint optimization processes. Furthermore, SGHC's final output is generated by concatenating semantic-quantized features with pixel-quantized features, preserving the full expressive capacity of the original semantic features while integrating complementary texture information through this unified feature fusion paradigm. Subsequent tasks, such as reconstruction, multimodal understanding, and generation, all share the same discrete token representation; (2) ***Efficient Downstream Applications:*** SGHC effectively avoids two critical predicaments: token quantity inflation and codebook overexpansion. As defined before, the semantic codebook size is $K$, and each pixel sub-codebook size is $m$. Due to dimensional concatenation, the complete codebook flattens to $K \times m$, where $m$ is much smaller than $K$. In our experimental default settings, we extend the complete codebook to a size comparable to existing LLMs' text vocabulary size, e.g., the size of Qwen2 vocabulary is 150k.

Table 2: Comparative analysis of tokenizers on multimodal comprehension tasks. *: Both the tokenizer and LLM are reproduced in our setting. Our method achieves SOTA performance compared with other discrete tokenizers.

| Model | LLM | Data | Res. | POPE | MME-P | SEED | GQA |
|---|---|---|---|---|---|---|---|
| SigLIP (Zhai et al., 2023) | Vicuna-7B | LLaVA-v1.5 | 256 | 83.76 | 1481.0 | 65.28 | 61.9 |
| LlamaGen (Sun et al., 2024a) | Vicuna-7B | LLaVA-v1.5 | 256 | 65.6 | 716.8 | 35.0 | 39.8 |
| VILA-U (Wu et al., 2024c) | Vicuna-7B | LLaVA-v1.5 | 256 | 81.6 | 1311.6 | 56.9 | 55.3 |
| SDE(MUSE-VL)* (Xie et al., 2025) | Vicuna-7B | LLaVA-v1.5 | 256 | 77.3 | 1240.0 | 56.7 | 58.0 |
| TokLIP (Lin et al., 2025) | Qwen2.5-7B-Ins | LLaVA-v1.5 | 256 | 81.2 | 1346.8 | 59.8 | 57.4 |
| **SemHiTok(Ours)** | Vicuna-7B | LLaVA-v1.5 | 256 | **82.5** | **1355.8** | **62.9** | **60.3** |
| TokenFlow-384 (Qu et al., 2024) | Vicuna-7B | LLaVA-v1.5 | 384 | 84.9 | 1416.4 | 62.7 | 61.2 |
| TokLIP (Lin et al., 2025) | Qwen2.5-7B-Ins | LLaVA-v1.5 | 384 | 82.7 | 1410.2 | **65.2** | 59.3 |
| **SemHiTok-384(Ours)** | Vicuna-7B | LLaVA-v1.5 | 384 | **86.3** | **1465.6** | 64.1 | **62.3** |

## 2.3 UNIFIED MLLM EQUIPPED WITH SEMHITOK

The framework diagram for unified MLLM is shown in Fig.2(c). We use SemHiTok to develop a unified multimodal model, which models discrete vision and text token sequences with a universal next-token prediction loss. Particularly, in image processing, SemHiTok is utilized to discretize images into token sequences. On the model side, we merely expand the text vocabulary and adjust the head layer to incorporate visual token IDs. To enable a unified head layer, we flatten SGHC by merging all sub-codebooks into a single flat representation as shown in Fig.2(b). Specifically, for the $j_{th}$ semantic code in the $i_{th}$ pixel sub-codebook, the discrete code index in the completed codebook is $h = i \times m + j$, where $m$ is the sub-codebook size. It is also worth noting that the vocabulary expansion is merely for implementation convenience. We still use the features extracted from SGHC as input and align with LLM through a lightweight adapter layer. After training is completed, we replace the visual component in the embedding layer in order to achieve consistency between training and inference. To enable LLM to better handle features at two different levels, we introduce a Dual-MLP adapter layer, which projects semantic features and pixel features separately, and then concatenates them along the dimension before feeding them into the LLM. To enable classifier-free guidance (Ho & Salimans, 2022), we randomly replace the text condition with a probability of 0.1 to the unconditioned text during training. More deployment details are provided in the **Supplementary Material 7.2**.

## 3 EXPERIMENTS

### 3.1 EXPERIMENTAL SETUP

**Tokenizer.** For the semantic branch, we employ SigLIP (Zhai et al., 2023) as the semantic encoder and three self-attention layers as the semantic decoder to reconstruct semantic features. For the pixel branch, we employ ViT as both the pixel encoder and decoder, assigning 8 pixel sub-codes to each semantic code. More tokenizer detail, please refer to **Supplementary Material 7.3**.

**Unified MLLM.** We use Qwen2.5-7B-Instruct (Yang et al., 2024) as the base LLM, and expand its vocabulary and output head layer. We evaluate visual understanding on standard VQA benchmarks including SEEDB (Li et al., 2023a), POPE (Li et al., 2023c), GQA (Hudson & Manning, 2019), MMMU (Yue et al., 2024), MMB (Liu et al., 2024b) and MME (Fu et al., 2023). For visual generation evaluation, we report results on MJHQ-30K (Li et al., 2024c), GenAI-Bench (Li et al., 2024a), GenEval (Ghosh et al., 2023) and DPG (Hu et al., 2024). More unified MLLM detail, please refer to **Supplementary Material 7.4**.

### 3.2 UNIFIED IMAGE TOKENIZER

**Image Reconstruction.** We present the reconstruction performance on the ImageNet-50k validation set in Tab.1. Notably, SemHiTok excels in reconstruction quality compared to the unified tokenizer, recording an impressive 1.16 rFID with $16\times$ downsampling ratio. While SemHiTok's vocabulary size(approx 196k) appears larger than baselines, it is crucial to consider the effective representational

Table 3: Quantitative results on multimodal understanding benchmarks. SemHiTok achieves SOTA performance on most benchmarks among *Und&Gen Discrete* MLLMs, and is comparable to or even surpasses some *Only Und* and *Und&Gen. Continuous* models. The performance on *Und&Gen Discrete* with top-1 and top-2 values is denoted in bold and underline, respectively.

| Method | # Params | Res. | SEED | POPE | GQA | MMMU | MMB | MME | MME-P | MMV |
|---|---|---|---|---|---|---|---|---|---|---|
| *Only Und.* | | | | | | | | | | |
| LLaVA-Phi (Zhu et al., 2024b) | 2.7B | 256 | - | 85.0 | - | - | 59.8 | - | 1335.1 | 28.9 |
| LLaVA-v1.5 (Liu et al., 2023b) | 7B | 336 | 58.6 | 85.9 | 62.0 | 35.4 | 64.3 | - | 1510.7 | 31.1 |
| Qwen-Vl-Chat (Bai et al., 2023b) | 7B | 448 | 57.7 | - | 57.5 | 30.5 | - | 1848.3 | 1487.5 | - |
| ShareGPT4V (Chen et al., 2024b) | 7B | 336 | 69.7 | - | 63.3 | 37.2 | 68.8 | 1943.8 | 1567.4 | 37.6 |
| *Und&Gen. Continuous* | | | | | | | | | | |
| LaVIT (Jin et al., 2023) | 7B | 224 | - | - | 46.8 | - | 58.0 | - | - | - |
| Janus (Wu et al., 2024a) | 1.5B | 384 | 63.7 | 87.0 | 59.1 | 30.5 | 69.4 | - | 1338.0 | 34.3 |
| Janus-Pro-1B (Chen et al., 2025) | 1.5B | 384 | 68.3 | 86.2 | 58.9 | 38.9 | 65.5 | - | 1444.0 | - |
| MAR (Wu et al., 2025) | 1.5B | 512 | 67.1 | 87.6 | 58.9 | 38.9 | 65.5 | - | 1155.0 | - |
| *Und&Gen. Discrete* | | | | | | | | | | |
| LWM (Liu et al., 2024a) | 7B | 256 | - | 75.2 | 44.8 | - | - | - | - | 9.6 |
| SEED-LLaMA (Li et al., 2024b) | 13B | 256 | 53.7 | - | - | - | - | - | - | - |
| Show-o (Xie et al., 2024) | 1.5B | 256 | - | 80.0 | - | 26.7 | - | - | 1097.2 | - |
| Liquid (Wu et al., 2024b) | 7B | 512 | - | 81.1 | **71.3** | - | - | - | 1119.3 | - |
| EMU3 (Wang et al., 2024c) | 8B | 512 | 68.2 | 85.2 | 60.3 | 31.6 | 58.5 | 1509.9 | 1243.8 | **37.2** |
| VILA-U (Wu et al., 2024c) | 7B | 256 | 56.3 | 83.9 | 58.3 | - | - | - | 1336.2 | 27.7 |
| VILA-U (Wu et al., 2024c) | 7B | 384 | 59.0 | **85.8** | 60.8 | - | - | - | 1401.8 | 33.5 |
| UniToken (Jiao et al., 2025) | 7B | 384 | 69.3 | - | - | 32.8 | 69.9 | - | - | - |
| TokLIP (Lin et al., 2025) | 7B | 384 | 76.9 | 84.1 | 59.5 | **43.1** | 67.6 | - | 1488.4 | 29.8 |
| TokenFlow-B (Qu et al., 2024) | 13B | 224 | 60.4 | 84.0 | 59.3 | 34.2 | 55.3 | 1660.4 | 1353.6 | 22.4 |
| TokenFlow-L (Qu et al., 2024) | 13B | 256 | 62.6 | 85.0 | 60.3 | 34.4 | 60.3 | 1622.9 | 1365.4 | 27.7 |
| SynerGen-VL (Li et al., 2025) | 2.4B | 512 | 62.0 | 85.3 | 59.7 | 34.2 | 53.7 | 1837.0 | 1381.0 | 34.5 |
| **SemHiTok(Ours)** | 7B | 256 | 69.7 | 83.4 | 60.3 | 39.3 | 72.3 | 1775.9 | 1449.0 | 30.5 |
| **SemHiTok(Ours)** | 7B | 384 | **79.8** | 85.5 | 61.7 | 41.0 | **75.2** | **1993.8** | **1512.8** | 36.6 |

Table 4: Comparison of generation quality on GenAI and MJHQ30K. SemHiTok achieves comparable results with specialist models and unified MLLMs.

| Model | Params | Type | #Training Images | Res. | GenAI-Bench | | MJHQ30K gFID ↓ |
|---|---|---|---|---|---|---|---|
| | | | | | Basic ↑ | Advanced ↑ | |
| *Only Gen.* | | | | | | | |
| SD v2.1 (Rombach et al., 2022) | – | Diff | 2000M | 1024 | 0.78 | 0.62 | – |
| DALL-E 3 (Betker et al., 2023) | – | Diff | - | 1024 | 0.90 | 0.70 | – |
| PixArt-α (Chen et al., 2023) | 0.6B | Diff | - | 1024 | – | – | 6.14 |
| SDXL (Podell et al., 2023) | 2.6B | Diff | 2000M | 1024 | 0.83 | 0.63 | 9.55 |
| Playgroundv2.5 (Li et al., 2024c) | – | Diff | - | 1024 | – | – | 4.48 |
| *Und&Gen.* | | | | | | | |
| LWM (Liu et al., 2024a) | 7B | AR | - | 256 | 0.63 | 0.53 | 17.77 |
| Show-o (Xie et al., 2024) | 1.5B | Diff | 36M | 256 | 0.70 | 0.60 | 15.18 |
| Janus (Wu et al., 2024a) | 1.3B | AR | 65M | 384 | – | – | 10.10 |
| VILA-U (Wu et al., 2024c) | 7B | AR | 15M | 256 | 0.76 | 0.64 | 12.81 |
| VILA-U (Wu et al., 2024c) | 7B | AR | 15M | 384 | 0.73 | 0.61 | 7.69 |
| SynerGen-VL (Li et al., 2025) | 2.4B | AR | 25M | 256 | - | - | 6.10 |
| SDE(MUSE-VL) (Xie et al., 2025) | 7B | AR | 10M | 256 | - | - | 7.73 |
| ILLUME (Wang et al., 2024a) | 7B | AR | 15M | 512 | 0.75 | 0.60 | 7.76 |
| ILLUME+ (Huang et al., 2025a) | 7B | AR | 15M | 512 | 0.75 | 0.60 | 7.76 |
| TokenFlow (Qu et al., 2024) | 7B | AR | 60M | 256 | 0.65 | 0.65 | 7.78 |
| Liquid (Wu et al., 2024b) | 7B | AR | 30M | 512 | **0.83** | 0.65 | 5.47 |
| UniTok (Ma et al., 2025) | 7B | AR | 30M | 256 | 0.85 | 0.67 | 7.46 |
| **SemHiTok(ours)** | 7B | AR | 15M | 256 | **0.83** | 0.64 | **5.40** |
| **SemHiTok(ours)** | 7B | AR | 15M | 384 | **0.83** | **0.66** | 5.70 |

capacity defined by the code shape. VILA-U(using RQ) and FQGAN(using Product Quantization) operate within a combinatorial search space($N^D$ or $N_1 \times N_2$), resulting in a significantly larger effective capacity than SemHiTok's linearly constrained structure($K \times m$). Furthermore, these approaches typically employ a denser code shape, whereas SemHiTok relies on a single unified hierarchical index. This indicates that our superior performance derives from the structured efficiency of SGHC rather than brute-force capacity expansion. Increasing the training 384 resolution led to a

Table 5: Impact of key design choices on reconstruction and multimodal understanding.

| Semantic Codebook | SGHC | Dual MLP | Enhance Decoder | MME-P↑ | MMB↑ | SEED↑ | MMU↑ | rFID↓ |
|:---:|:---:|:---:|:---:|:---:|:---:|:---:|:---:|:---:|
| ✓ | | | | 1387.5 | 61.3 | 62.3 | 35.6 | 3.17 |
| ✓ | ✓ | | | 1355.8 | 60.7 | 62.9 | 35.8 | 1.42 |
| ✓ | ✓ | ✓ | | 1393.0 | 61.6 | 63.2 | 36.1 | 1.42 |
| ✓ | ✓ | ✓ | ✓ | 1393.0 | 61.6 | 63.2 | 36.1 | 1.16 |

significant improvement in the rFID score, reaching 0.66. The results validate the effectiveness of SGHC design in modeling pixel feature space of the semantic code.

**LLaVa-v1.5 Multimodal Understanding.** To ensure a fair comparison, we conduct experiments to evaluate the multimodal understanding performance of existing open-source tokenizers and SemHiTok under the standard LLaVA-v1.5 setting. The results are as shown in Tab.2. LlamaGen's performance is notably inferior, a consequence of its deficient pre-alignment with text. In contrast, unified image tokenizers show considerable advancements in understanding. However, due to their inherent hybrid structures or joint training strategies, a substantial disparity in understanding performance remains between prior discrete tokenizers and continuous representations. Notably, even though TokLIP uses a stronger base model (Qwen2.5-7B-Ins), its performance remains inferior to ours. SemHiTok achieves state-of-the-art results for discrete tokenizers, nearing the performance of continuous inputs such as SigLIP.

## 3.3 UNIFIED MLLM

**Multimodal Understanding.** We evaluate the understanding performance of SemHiTok on diverse benchmarks in Table 3. *Und* means support multimodal understanding, *Gen* means support text-to-image generation. Among *Und&Gen. Discrete*, SemHiTok achieves state-of-the-art performance on most metrics, such as SEED, MMB, MME, and MME-P. Compared to other unified tokenizers (e.g., VILA-U, TokenFlow, UniToken), our models demonstrate significant advantages. Notably, our model surpasses expert-level models on key benchmarks, achieving 3.8 and 6.4 points higher than ShareGPT4V on MMMU and MMB, respectively. This bridges the gap between discrete visual tokens and continuous visual tokens in multimodal understanding tasks, strongly demonstrating the superiority and potential of our approach. Visualizations on understanding tasks are available in **Supplementary Material 7.5**.

**Text-to-Image Generation** To evaluate the text-to-image generation, we use GenAI-bench (Li et al., 2024a) and MJHQ30K (Li et al., 2024c) benchmark, and the results are shown in Tab4. For GenAI-bench, we use clip-flant5-xxl as the VQA score model to reflect the consistency between text descriptions and generated images. On this challenging benchmark, our model achieves competitive performance, closely matching Liquid, even though Liquid employs a generation-focused tokenizer. Furthermore, our model even outperformed some diffusion-based expert models, such as SDXL and SD v2. The strong results underscore the superior capability of our unified MLLM in complex text-to-image generation tasks. For MJHQ30K, we use the generation FID metric on generated images and high-quality images. On this benchmark, SemHiTok-256 attains 5.40 gFID, setting a new state-of-the-art in autoregressive image generation. In addition, we conduct further quantitative comparisons on GenEval and DPG, which are available in **Supplementary Material 7.6**. More quantitative analysis results and visualizations on generation tasks are available in **Supplementary Material 7.7**.

## 4 ABLATION

### 4.1 IMPACT OF KEY DESIGN.

In Tab.5, we validate the impact of our key design choices in SemHiTok: semantic codebook, semantic-guided hierarchical codebook(SGHC), Dual MLP, and Enhance Deocoder. For efficiency, we only test the understanding performance under LLaVA-v1.5, and train 40 epochs on ImageNet-1K for reconstruction tasks. We begin with the semantic codebook, which suffices for multimodal

Table 6: Impact of training strategy and structure. **DStruct**: Using different learnable codebooks to model semantic and pixel information; **DTrain**: Adopt a phased optimization training strategy. **Exp 1**: used the same architecture as SDE and adopted a joint training strategy; **Exp 2**: using the same architecture as SemHiTok but with joint training; **Exp 3**: using two separately pretrained tokenizers (Llamagen and SigLIP) to independently extract semantic and pixel information.

| Exp | DStruct | DTrain | Multimodal understanding | | | | Reconstruction | |
|-----|---------|--------|-------|--------|--------|-------|-------|--------|
| | | | GQA↑ | MME-P↑ | SEEDB↑ | POPE↑ | rFID↓ | Usage↑ |
| 1 | ✗ | ✗ | 58.0 | 1240.0 | 56.7 | 77.3 | 3.78 | 92.9% |
| 2 | ✓ | ✗ | 57.8 | 1357.4 | 55.3 | 80.4 | 3.22 | 45.9% |
| 3 | ✓ | ✓ | 58.7 | 1210.9 | 56.1 | 80.1 | 2.19 | 97.0% |
| 4 | SemHiTok | | 60.3 | 1355.8 | 62.9 | 82.6 | 1.42 | 93.7% |

understanding but suffers from poor reconstruction. Incorporating SGHC enables pixel reconstruction and reduces rFID by 1.75, without noticeably affecting understanding performance. Introducing Dual-MLP further enhances multimodal understanding, even surpassing the semantic codebook alone, highlighting the effectiveness of multi-level feature modeling. Finally, a stronger pixel decoder brings additional improvements in reconstruction quality. figure 5 demonstrates the effects of various modules on reconstruction results. Compared to the semantic codebook, SGHC delivers more detailed reconstructions. Additionally, the enhanced decoder boosts performance on difficult samples. More reconstruction comparisons are available in **Supplementary Material 7.8**.

## 4.2 IMPACT OF TRAINING STRATEGY AND STRUCTURE.

We conduct comparative experiments under the same hyperparameters to investigate the impact of the training strategy and structure, as shown in Table 6. It can be observed that **Exp 1**'s performance is the lowest across both multimodal understanding and reconstruction tasks. After decoupling the architecture, **Exp 2** achieves superior performance compared to **Exp 1**, yet its codebook usage remains at a low level. Furthermore, **Exp 4** achieves marked gains in both efficacy and utilization over **Exp 2**, which underscores the importance of a phased training strategy for the proposed SemHiTok hierarchical architecture. Compared with **Exp 3**, **Exp 4** achieves superior performance on both multimodal understanding and reconstruction tasks. This suggests that naively incorporating pixel features may negatively affect the alignment between image features and the LLM. In contrast, **Exp 4** leverages pixel features as a complementary refinement to semantic features, thereby effectively bridging the gap between semantic and pixel representations.

## 4.3 MORE COMPARISONS AND ABLATION STUDIES

In **Supplementary Material 7.9**, we attempt to provide quantitative evidence to further demonstrate pixel similarity within the semantic code. Meanwhile, to further illustrate how different hyperparameters in our method affect model performance, we also conduct ablation studies on Concat Type and Sub-Codebook Size in **Supplementary Material 7.10**, Semantic VQ Type, Codebook dim and Codebook size in **Supplementary Material 7.11**, and different $K \times m$ and more comparison with other large codebook reconstruction methods in **Supplementary Material 7.12**.

## 5 RELATED WORK

### 5.1 SPECIALIZED IMAGE TOKENIZER

**Tokenization for Autoregressive Generation.** VQVAE(Van Den Oord et al., 2017) learns a discrete representation using a learnable codebook in auto-encoder architectures. VQGAN further improves the perceptual quality by using adversarial training(Goodfellow et al., 2014). improved the architecture with perceptual and adversarial-based losses for better reconstruction. This approach yields more precise and detailed image representations, significantly improving upon previous methodologies in image generation and processing. In recent literature, researchers are turning to efficient codebook structures(Shi et al., 2024; Zhang et al., 2023; Yu et al., 2021b; Bai et al., 2024) and better quantization

methods(Zha et al., 2024; Yu et al., 2025) to improve generation performance and compression rates. IBQ(Shi et al., 2024) proposes the Index Backpropagation Quantization codebook update method, achieving stable training of large-scale codebooks. FQGAN(Bai et al., 2024) uses multiple codebooks and product quantization, where each codebook encodes a different type of feature. However, this work focuses only on image reconstruction and generation tasks, performing poorly on multimodal understanding tasks(Wu et al., 2024b; Liu et al., 2024a).

**Tokenization for Understanding.** In multimodal large language models (MLLMs)(Li et al., 2023b; 2022; Radford et al., 2021b; Liu et al., 2023a; Bai et al., 2023a; Chen et al., 2024c), researchers leverage CLIP(Radford et al., 2021b) and BLIP(Li et al., 2022) to extract visual characteristics that align with the language during its pre-training phase. Building upon many works (Liu et al., 2023a; Bai et al., 2023a; Chen et al., 2024c) have been collected and trained on high-quality datasets to achieve remarkable performance. LLaVA(Liu et al., 2023a) utilizes a vision encoder to align the vision inputs before LLMs. QwenVL(Bai et al., 2023a) and InterVL(Chen et al., 2024c) achieve better results through increased resolution, higher-quality datasets, etc. However, these text-aligned image encoders tend to focus on semantic information and ignore texture information, which is important for the generation task.

## 5.2 UNIFIED IMAGE TOKENIZER

Numerous efforts have emerged to develop unified visual generation and understanding within one MLLM(Wang et al., 2024a;c; Xie et al., 2024; Dong et al., 2023; Ge et al., 2024; Sun et al., 2024b; Team, 2024; Wu et al., 2024d). There are two main lines to bridge the gap. Many workers (Dong et al., 2023; Ge et al., 2024; Sun et al., 2024b) combine diffusion models with LLMs for image generation. DREAMLLM(Dong et al., 2023) presents a unified framework that not only provides multimodal understanding but also creates multimodal content via diffusion models. Emu2(Sun et al., 2024b) trains a unified generative model using a diffusion-based decoder. These approaches inevitably increase model complexity and are not simple enough. Other workers (Team, 2024; Wu et al., 2024d; Xie et al., 2024; Wang et al., 2024c) adopt VQVAE-based encoders to convert images into discrete tokens. Chameleon(Team, 2024) and EMU3(Wang et al., 2024c) directly use VQGAN(Yu et al., 2021a), which is optimized by pixel reconstruction as the image tokenizer, while this method increases resource consumption during the pre-training stage and degrades multimodal understanding performance. VILA-U(Wu et al., 2024d) introduces a unified image tokenizer that incorporates a text-aligned branch within the VQGAN training paradigm. However, due to the gap between the semantic feature and the pixel feature, the joint optimization approach may lead to suboptimal solutions In contrast, our proposed SemHiTok can add the ability of extracting texture features without changing the discrete semantic features, and avoids the challenges brought by joint optimization.

## 6 CONCLUSION

**Conclusion:** We present SemHiTok, a unified image tokenizer featuring a Semantic-Guided Hierarchical Codebook (SGHC) to balance semantic depth with reconstruction fidelity. SemHiTok achieves SOTA results on LLaVA-v1.5 understanding and ImageNet-50k reconstruction benchmarks. When integrated into a unified MLLM, it delivers competitive performance in both understanding and generation, offering a powerful, discrete, and fully compatible alternative to existing tokenizers.

**Limitation:** We present two limitations: *(1). Low generation efficiency:* Owing to the use of standard image quantization methods and settings, each 256-resolution image is represented by 256 tokens. This results in relatively low efficiency and high computational cost. *(2). Unified large model potential:* In tasks involving natural language understanding and multimodal reasoning, advanced post-training techniques such as Chain-of-Thought (CoT) are not explored. This remains a promising direction for future research in the community.

**Future Work:** In the future, we can explore the potential of unified image tokenizers and test their performance on more difficult tasks, such as image editing and multiple rounds of conversations. In addition, improving the compression ratio of the model and designing a tokenizer that serializes the image in one dimension are also expected.

ACKNOWLEDGEMENT

This work is supported by National Key Research and Development Program of China(2024YFE0203100), Scientific Research Innovation Capability Support Project for Young Faculty (No.ZYGXQNJSKYCXNLZCXM-I28), National Natural Science Foundation of China (NSFC) under Grants No.62476293 and No.62372482, and General Embodied AI Center of Sun Yat-sen University.

ETHICS STATEMENT

This research does not involve potentially harmful insights, methodologies, or applications, and it raises no concerns regarding conflicts of interest, sponsorship, discrimination, bias, fairness, privacy, security, legal compliance, or research integrity.

REPRODUCIBILITY STATEMENT

**Data.** The training datasets employed in this study are detailed in Appendix.7.3 and Appendix. 7.4, and all are publicly accessible open-source resources.

**Method.** To support reproducibility, we elaborate on the methodological details in Section.2, and the implementation will be open-sourced after acceptance.

**Performance.** All evaluations are carried out on open benchmarks, thereby ensuring the reproducibility of our results.

**Code And Model Weight.** We will open-source the code and model weight files.

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

# 7 TECHNICAL APPENDICES AND SUPPLEMENTARY MATERIAL

## 7.1 LLM USAGE STATEMENT

We clarify that the use of LLMs in this study is restricted to writing assistance, specifically for grammar correction and enhancing readability. No LLM was involved in the research design, experimental execution, or data analysis.

## 7.2 DEPLOYMENT DETAILS OF UNIFIED MLLM.

**Training Data Form.** For multimodal understanding, we use <|im_start|> and <|im_end|> to delimit the image segment within the input sequence. To distinguish between modalities and enable visual content generation, we insert special tokens: <IMG_XXXXX>, which represent image codes in LLMs' vocabulary. Specifically, the i-th code in the unified tokenizer codebook corresponds to <IMG_i>. In addition, we add <start_of_image> and <end_of_image> to indicate the start and end of image generation.

**Vocabulary Embedding Processing Flow.** For understanding samples, we follow the LLaVA setting: discrete features are first extracted by the tokenizer and then fed into the LLM through an adapter layer. To ensure consistency between understanding and generation, we employ the same features for generation samples instead of using the embeddings of <IMG_XXXXX>. After training, we replace the visual code embedding in the vocabulary to align training with inference. The detailed procedure is illustrated in Algorithm 1.

---

**Algorithm 1:** Procedure: Training and Generation Pipeline

---

**Before Training:**;

1. Add special tokens <IMG_XXXXX> into LLM vocabulary 2. Preprocess training images, e.g.
 "generate a dog + `<Image>`" →
 "generate a dog + `<start_of_image><IMG_i>...<IMG_k><end_of_image>`"

**During Training:**;

1. Feed preprocessed samples into LLM tokenizer → `text_id`;
2. Lookup embedding $V$ from `text_id` → $E_{text}$;
3. Get image IDs from `text_id` → `img_id`;;
Lookup unified codebook embedding from `img_id` → $E_{img}$;;
Apply adapter layer: $E_{img} \rightarrow E'_{img}$;
4. Replace image embeddings in $E_{text}$ with $E'_{img} \rightarrow E'_{text}$;
5. Feed $E'_{text}$ into LLM backbone for training;

**Generation:**;

1. Apply adapter layer on codebook embeddings $C_{img} \rightarrow C'_{img}$;
2. Replace image part of $V$ with $C'_{img} \rightarrow V'$;
3. Start autoregressive generation;

---

## 7.3 TOKENIZER EXPERIMENTAL DETAILS

The training of SemHiTok is conducted in two stages. During semantic codebook training, we train the semantic tokenizer for one epoch on 50M subset of COYO-700M(Byeon et al., 2022). For the PRE stage, we first train the model(ViT-Base) on ImageNet(Deng et al., 2009), and then fine-tune on 50M COYO to improve its generalization, following LlamaGen(Sun et al., 2024a). To further improve the reconstruction and generation performance, we enlarge the size of the pixel decoder(ViT-Large) and fine-tune on the 20M COYO data and 20M MidJourney-style synthetic data. The full training takes about 3 days on 32 v100 GPUs.

## 7.4 UNIFIED MLLM EXPERIMENTAL DETAILS

Following existing work(Wu et al., 2024c; Ma et al., 2025), we first pretrain the model and adapter layer on a mix of multimodal data, which is composed of 3.5M language data from Magpie(Xu et al.,

2024b) and Openorca(Lian et al., 2023), 10M caption image-text pairs data, and 15M MidJourney-style synthetic data. Subsequently, we finetune the model on 1M language dataset from Magpie(Xu et al., 2024b) and Evol-Instruct(Xu et al., 2024a), 4M generation data and 4M understanding data from emova(Chen et al., 2024a) and LLaVA-SFT(Liu et al., 2023b). The full training takes about 7 days on 32 A800 GPUs. During image generation inference, we apply classifier-free guidance(Ho & Salimans, 2022) with a scale factor of 2.5.

## 7.5 VISUALIZATIONS ON UNDERSTANDING TASKS

We present more visualization of multimodal understanding samples in figure 6.

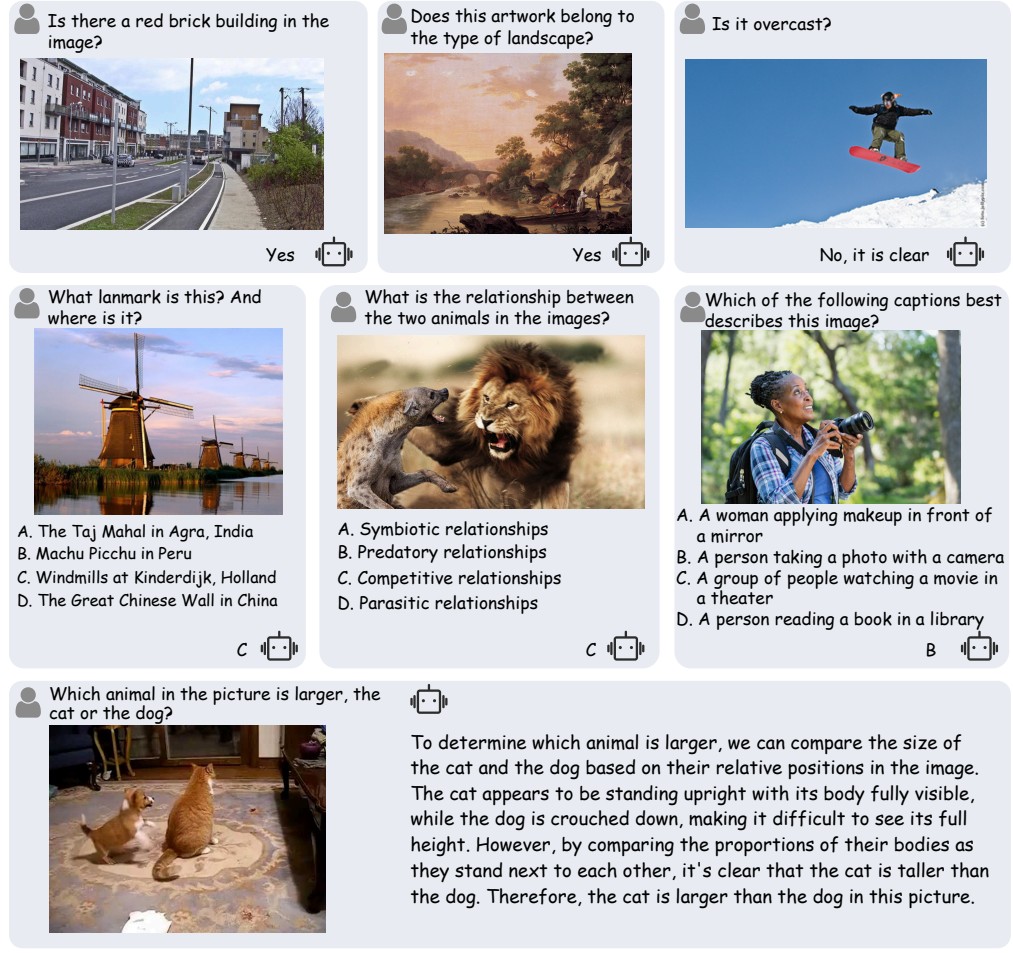

Figure 6: Visualizations on understanding tasks.

## 7.6 MORE COMPARISON OF GENERATION ON GENEVAL AND DPG

To more fully demonstrate the superiority of SemHiTok, we conduct further comparisons with other generative models and unified models on GenEval Ghosh et al. (2023) and DPG Hu et al. (2024). As shown in Tab.7, our method still demonstrates competitive performance.

## 7.7 VISUALIZATIONS ON GENERATION TASKS

We present more visualization of generated images in figure 7.

Table 7: Comparison with other methods on GenEval and DPG Bench. SemHiTok still achieves competitive performance even with a smaller amount of data.

| Model | Params | Type | #Training Images | Res. | GenEval | DPG |
|---|---|---|---|---|---|---|
| *Only Gen.* | | | | | | |
| SD v2.1 (Rombach et al., 2022) | – | Diff | 2000M | 1024 | 0.50 | – |
| DALL-E 2 (Ramesh et al., 2022) | – | Diff | 650M | 1024 | 0.52 | – |
| DALL-E 3 (Betker et al., 2023) | – | Diff | – | 1024 | 0.67 | 83.50 |
| PixArt-$\alpha$ (Chen et al., 2023) | 0.6B | Diff | – | 1024 | 0.48 | – |
| SDXL (Podell et al., 2023) | 2.6B | Diff | 2000M | 1024 | 0.55 | 74.65 |
| Playgroundv2.5 (Li et al., 2024c) | – | Diff | – | 1024 | – | 75.47 |
| *Und&Gen.* | | | | | | |
| Transfusion (Zhou et al., 2024) | 7.3B | AR | 3.5B | 256 | 0.63 | – |
| Ming-Lite-Uni (AI et al., 2025) | 8B | AR-Scale | 5M | 512 | 0.62 | – |
| Show-o (Xie et al., 2024) | 1.5B | Diff | 36M | 256 | 0.53 | 67.27 |
| Emu3 (Wang et al., 2024c) | 8B | AR | – | 256 | 0.66 | 80.60 |
| Janus (Wu et al., 2024a) | 1.3B | AR | 65M | 384 | 0.61 | – |
| SynerGen-VL (Li et al., 2025) | 2.4B | AR | 25M | 256 | 0.61 | – |
| UniFork (Li et al., 2025) | 0.76B | AR | – | 384 | 0.46 | – |
| ILLUME (Wang et al., 2024a) | 7B | AR | 15M | 512 | 0.61 | – |
| ILLUME+ (Huang et al., 2025a) | 3B | AR | 46M | 512 | 0.72 | – |
| MUSE-VL (Xie et al., 2025) | 7B | AR | 10M | 256 | 0.57 | – |
| QLIP-B (Zhao et al., 2025) | 1.5B | AR | 18M | 256 | 0.48 | 78.17 |
| TokenFlow (Qu et al., 2024) | 7B | AR | 60M | 256 | 0.63 | 73.38 |
| MMaDA (Yang et al., 2025) | 8B | Diff | – | 512 | 0.63 | 69.97 |
| Liquid (Wu et al., 2024b) | 7B | AR | 30M | 512 | 0.55 | 83.45 |
| UniTok (Ma et al., 2025) | 7B | AR | 30M | 256 | 0.59 | 83.45 |
| Tar-1.5B (Han et al., 2025) | 3B | AR | 46M | 256 | 0.76 | 82.96 |
| Tar-1.5B W/Self-Reflect (Han et al., 2025) | 3B | AR | 46M | 256 | 0.78 | 84.10 |
| Tar-7B (Han et al., 2025) | 7B | AR | 46M | 256 | 0.84 | 84.19 |
| Tar-7B W/Self-Reflect (Han et al., 2025) | 7B | AR | 46M | 256 | 0.85 | 84.65 |
| Ming-UniVision (Huang et al., 2025b) | 16B-A3B | AR-Continuous | – | 512 | 0.85 | 82.12 |
| **SemHiTok(ours)** | 7B | AR | 15M | 256 | 0.71 | 83.59 |

Table 8: Quantitative evaluation of pixel consistency using Variance Reduction Ratio (VRR). SemHiTok$_{sem}$ significantly outperforms the random baseline, verifying the intrinsic semantic-pixel correlation. Furthermore, our hierarchical refinement (SemHiTok$_{pix}$) achieves the best consistency, exceeding the standard VQGAN baseline.

| Experiment | Random Baseline | SemHiTok$_{sem}$ | VQGAN | SemHiTok$_{pix}$ |
|---|---|---|---|---|
| VRR | 0.185%±0.005% | 5.245%±0.085% | 9.535%±0.095% | 13.705%±0.075% |

## 7.8 MORE VISUALIZED RECONSTRUCTION RESULTS FROM THE ABLATION OF KEY MODULE

We show more reconstruction effects on the ablation of key modules in figure 8.

## 7.9 QUANTITATIVE ANALYSIS OF PIXEL FEATURES IN THE SEMANTIC CODEBOOK

In Fig.4, we provide visualization results showing that image patches corresponding to the same semantic code share similar pixel features. To further support this analysis, we attempt to provide quantitative evidence using the following method:

1. Using the tokenizer to extract the code indices along with their corresponding image patches.
2. Applying the **DCT(Discrete Cosine Transform)** (Rao & Yip, 2014) to extract patch's pixel features.
3. For each code, we compute the variance of the DCT features extracted from its associated image patches, followed by **averaging these variances over all codes**, denoted as $V_{mean}$. The more similar the pixels of the tokenized results are, the smaller the $V_{mean}$ becomes.
4. Computing the **global variance of all image patches**, denoted as $V_{global}$.

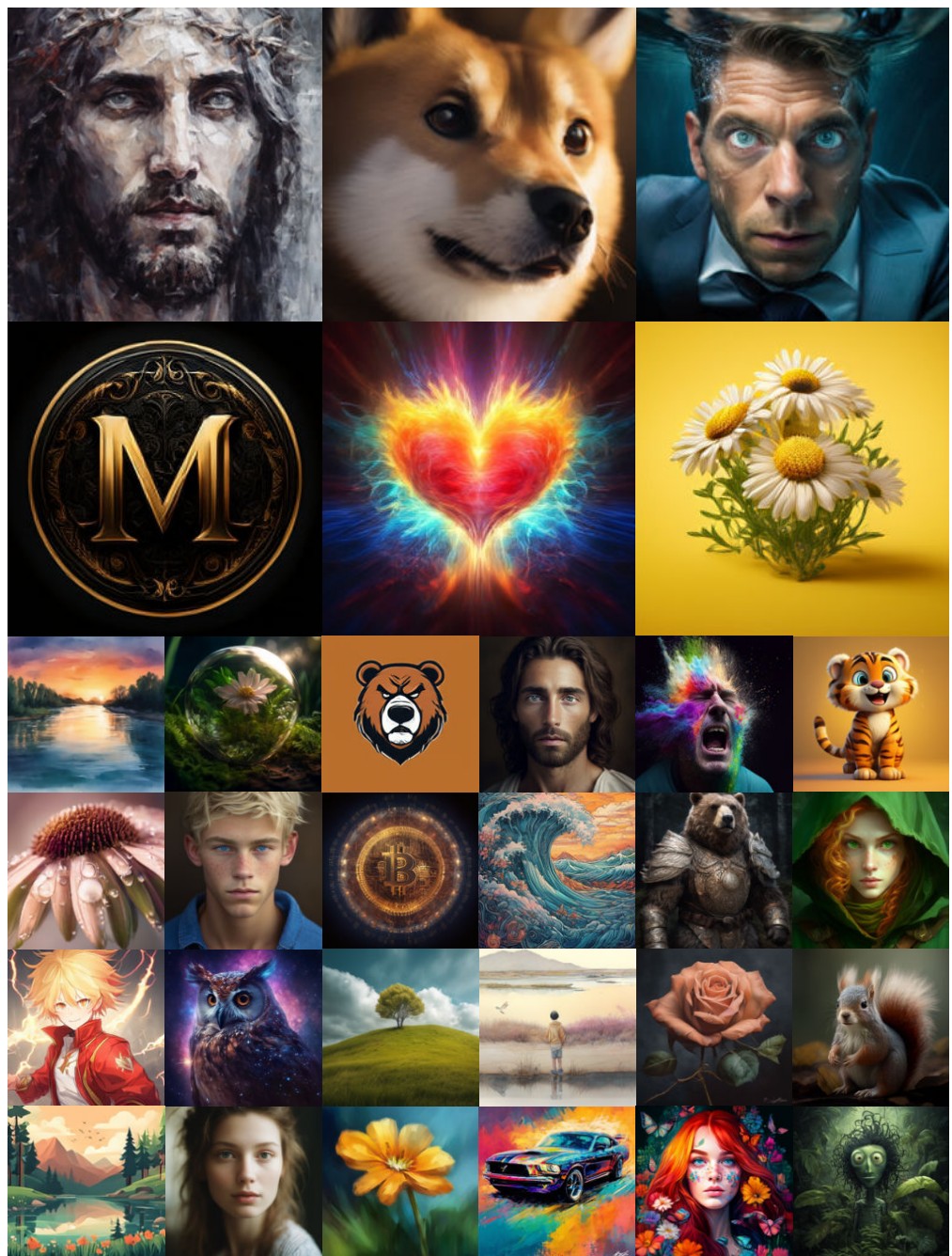

Figure 7: More generated images presentation.

5. Computing **Variance Reduction Ratio (VRR)**, denoted as $VRR = 1 - V_{mean}/V_{global}$. The larger VRR indicates that the patches corresponding to the given code have more similar pixel features.

We randomly sampled 10K images from the ImageNet-Val (50K) dataset across 5 independent runs. The Random Baseline refers to randomly tokenized results, serving as the lower bound for the VRR metric. VQGAN (Yu et al., 2021a), as an expert model for pixel reconstruction, serves as the standard reference for the VRR metric.

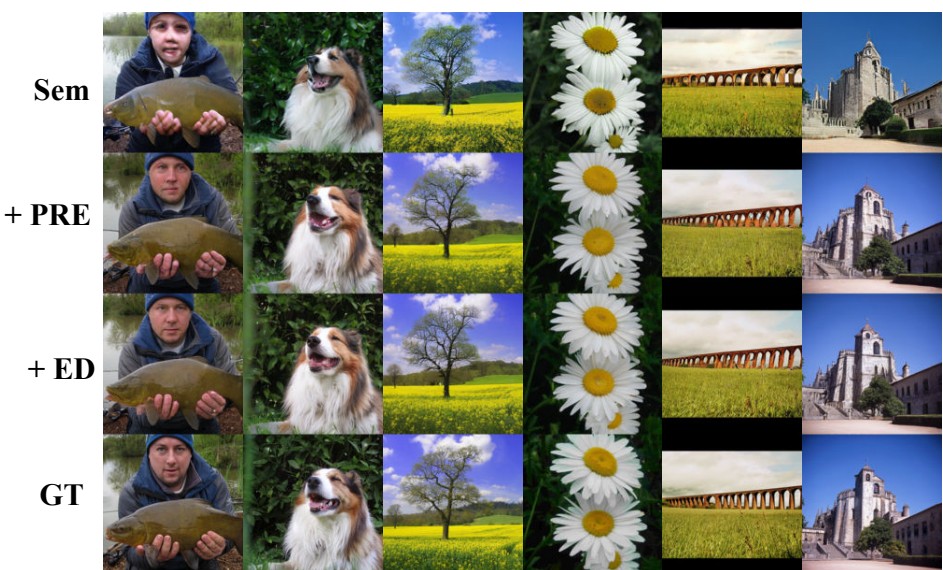

Figure 8: More visualized reconstruction results from the ablation of key module.

As illustrated in Tab. 8, the results provide strong quantitative evidence for the effectiveness of our Semantic-Guided Hierarchical Codebook (SGHC) design. **1.)Validation of Semantic-Pixel Correlation.** The SemHiTok$_{sem}$ achieves a VRR of 5.245%, which is significantly higher than the Random Baseline (0.185%). This quantitatively verifies our observation in Subsec2.2 that image patches assigned to the same semantic code naturally exhibit intrinsic pixel-level similarities. This correlation forms the theoretical foundation for using semantic codes to guide pixel quantization. **2.)The Necessity of Pixel Branch.** However, compared to VQGAN (9.535%), the lower VRR of SemHiTok$_{sem}$ confirms that semantic codes alone are insufficient to capture fine-grained high-frequency details. This justifies the necessity of our pixel branch design to bridge the gap between semantic abstraction and pixel fidelity. **3.)Superiority of Hierarchical Design.** Most notably, SemHiTok$_{pix}$ achieves the highest VRR of **13.705%**, surpassing the expert pixel reconstruction model VQGAN by a substantial margin (+4.17%). This result demonstrates that decomposing the complex global pixel space into semantic-conditioned sub-spaces allows for significantly tighter feature clustering and lower variance than standard global quantization methods. This explains why SemHiTok achieves superior reconstruction performance despite utilizing a decoupled training strategy.

### 7.10   IMPACT OF CONCAT TYPE AND SUB-CODEBOOK SIZE.

For efficiency, we only conduct training and evaluation on ImageNet-1K. We investigate the impact of the concat type of semantics between pixel and the sub-codebook size, on reconstruction performance

Table 9: Impact of Concat type and sub-codebook size. w/o sem: not use the semantic discrete token. The gray bar represents the default setting in our experiments.

| Concat | Subc Size | rFID ↓ | Usage↑ |
|--------|-----------|--------|--------|
| w/o | 12 | 1.99 | 95.4% |
| Len | 12 | 1.45 | 94.1% |
| Dim | 8 | 1.42 | 96.4% |
| Dim | 12 | 1.26 | 93.7% |
| Dim | 16 | 1.19 | 79.3% |

Table 10: Impact of VQ Type and dim of the semantic codebook. We evaluate multimodal understanding performance under the LLaVA-v1.5 setting. The gray bar represents the default setting in our experiments.

| VQ Type | Codebok Dim | MME-P↑ | MMB↑ | SEED↑ | MMU↑ |
|---------|-------------|--------|------|-------|------|
| Norm | 48 | 1249.6 | 52.2 | 52.8 | 34.8 |
| Vanilla | 48 | 1319.9 | 56.3 | 57.2 | 33.1 |
| EMA | 32 | 1385.9 | 58.3 | 61.2 | 35.1 |
| EMA | 48 | 1387.5 | 61.3 | 62.3 | 35.6 |
| EMA | 64 | 1428.7 | 60.9 | 62.5 | 35.5 |

Table 11: Ablation of Semantic Codebook K. We evaluate on multimodal understanding tasks.

| $K$ | POPE | MME-P | SEED | GQA | Usage |
|---|---|---|---|---|---|
| 4096 | 79.7 | 1297.7 | 60.4 | 56.6 | 100 |
| 8192 | 81.9 | 1343.0 | 61.7 | 59.7 | 99.7 |
| **16384** | 82.5 | 1355.8 | 62.9 | 60.3 | 99.0 |
| 32768 | 82.8 | 1364.2 | 62.5 | 60.6 | 93.0 |

Table 12: Comparative experiments with Different $K \times m$ on reconstruction task.

| $K$ | $m$ | rFID↓ | Usage↑ |
|---|---|---|---|
| 8192 | 12 | 1.58 | 95.5 |
| | 16 | 1.45 | 92.7 |
| | 24 | 1.31 | 89.8 |
| | 32 | 1.22 | 83.7 |
| | 8 | 1.42 | 96.4 |
| **16384** | 12 | 1.26 | 93.7 |
| | 16 | 1.19 | 79.3 |

Table 13: Quantitative comparison of reconstruction quality under consistent total codebook sizes. SemHiTok outperforms existing quantization methods at the 65k scale and achieves comparable rFID to expert models at the 262k scale.

| Method | Codebook Size | rFID↓ | Usage↑ |
|---|---|---|---|
| VQGAN-LC (Zhu et al., 2024a) | 65546 | 2.63 | 100.0 |
| VQGAN-LC(CLIP) (Zhu et al., 2024a) | 65546 | 2.40 | 100.0 |
| FSQ (Bai et al., 2024) | 64000 | 2.80 | 100.0 |
| LFQ (Yu et al., 2023) | 65536 | 2.88 | 100.0 |
| SimVQ (Zhu et al., 2025) | 65536 | 2.24 | 100.0 |
| **SemHiTok** | 8192×8(65536) | 1.93 | 98.1 |
| | 16384×4(65536) | 1.84 | 99.0 |
| SimVQ (Zhu et al., 2025) | 262144 | 1.99 | 100.0 |
| Open-MAGVIT2 (Luo et al., 2024) | 262144 | 1.17 | 100.0 |
| IBQ (Shi et al., 2024) | 262144 | 1.00 | 79.3 |
| **SemHiTok** | 8192×32(262144) | 1.21 | 83.7 |
| | 16384×8(262144) | 1.19 | 79.3 |

as shown in Tab. 9. For the Concat type, w/o semantic tokens leads to a significant drop in reconstruction quality, with score increase of 0.73 compared to the default setting. Furthermore, concatenation along the sequence length performs worse than along the dimension, as the two token sets are spatially aligned, making dimensional concatenation more appropriate. For sub-codebook size, increasing the size can improve the model's reconstruction performance, but it exhibits marginal utility. In addition, the codebook usage significantly decreases when the sub-codebook size is set to 16, which indicates that too large a sub-codebook size is not cost-effective.

## 7.11 IMPACT OF SEMANITC VQ TYPE, CODEBOOK DIM AND CODEBOOK SIZE.

In Tab.10, we ablate VQ type and codebook dimension on multimodal understanding. Specifically, Norm VQ is not suitable for semantic discretization and shows the worst performance in understanding tasks. This indicates that it is difficult to discretely model complex and rich semantic information by Norm VQ. Replacing it with vanilla VQ brings a clear improvement. To further stabilize the semantic codebook, we use EMA VQ, which achieves the best results. For the semantic codebook dimension, higher values offer better representation but with a marginal effect. We empirically set the dimension to 48 as the default.

Furthermore, we investigate the impact of semantic codebook size $K$ on multimodal understanding in Table 11. The results indicate that while larger codebooks generally enhance performance, increasing $K$ beyond 16,384 yields only marginal gains and leads to a noticeable decline in codebook usage. Therefore, we select $K = 16,384$ as the optimal balance between representational capacity and utilization efficiency.

### 7.12 Impact of different $K \times m$ and more comparison with other large codebook reconstruction methods.

We investigate the impact of semantic codebook size $K$ and sub-codebook size $m$ in Table 12. The results indicate that increasing either $K$ or $m$ generally enhances reconstruction fidelity (lower rFID). However, excessive expansion leads to a significant decline in codebook usage. Furthermore, when controlling for the total codebook size ($K \times m$), we observe that prioritizing a larger semantic codebook $K$ yields slightly better performance than increasing the sub-codebook size $m$.

To further demonstrate the effectiveness of our method, we conduct a more detailed comparison with other approaches that use large codebooks. As show in Tab.13, we conduct comparisons on the reconstruction task under two codebook sizes (65K and 262K). Under the 65K setting, our model surpasses the expert reconstruction model while maintaining good usage. Under the 262K setting, our method still achieves competitive performance compared to the expert model.

