# OpenReview forum: "SemHiTok: A Unified Image Tokenizer via Semantic-Guided Hierarchical Codebook for Multimodal Understanding and Generation"
_ICLR.cc/2026/Conference — ICLR 2026 Poster_

### Official Review · Reviewer_bysp · 2025-10-29

**Soundness:** 3
**Presentation:** 3
**Contribution:** 2
**Rating:** 4
**Confidence:** 4

**Summary:**

This paper introduces a unified tokenzier for both image understanding and generation. By linking a set of pixel code with a semantic code, the proposed SemHiTok makes a good trade-off between semantic image understanding and pixel image generation. Experiments on image reconstruction, understanding and generation confirm its effectiveness.

**Strengths:**

- The motivation and illustration of the proposed method are clear and easy to follow.
- Experiment results are good on multiple tasks, including image reconstruction, understanding and generation.
- The ablation experiments are clear to demonstrate the effect of each component.

**Weaknesses:**

- Image reconstruction. The rFID is not enough to prove the real performance of a tokenizer for image reconstruction. Other metrics like PSNR are encouraged. Besides, unified tokenizers like UniTok, MUSE-VL should be compared in Tab.1.
- Training setting. Most unified models are jointly trained on a mixture of multimodal understanding and text-to-image data, the proposed method are only trained on LLaVA-1.5 and text-to-image settings seperately, which may not reflect the relation between image understanding and generation under a unified model.
- Generation benchmarks. Recent popular benchmarks such as Gen-Eval and DPG are missing. Besides, it's better to include more recent methods on Tab.4.

**Questions:**

no.

---

> ### Author Response · Authors · 2025-11-24
> **Reply to Reviewer bysp**
>
> We thank reviewer bysp for your insightful comments. We are glad that you recognize our method is **“clear and easy to follow”**, and appreciate the comprehensive experimental results. In the following, we address all concerns and questions:
>
> > **W1: More reconstruction performance**
>
> **A1**: We appreciate the reviewer’s concern regarding the reconstruction performance. We **present the reconstruction metrics of several tokenizers in Table R1**, including rFID, PSNR, and SSIM. Our model still achieves competitive performance. It is worth noting that **the SDE in the initial version refers to the tokenizer in MUSE-VL**, and we will also add this in the latest version of the paper.
>
> | Method      | Codebook Size | Code Shape  | rFID$\downarrow$ | PSNR$\uparrow$ | SSIM$\uparrow$ |
> | ---         | :---:         | :---:       | :---:| :---:| :---:|
> | LlamaGen    | 16384         | 16x16       | 2.47 | 20.65| 0.54 |
> | Show-o      | 8192          | 16x16       | 3.21 | 21.34| 0.59 |
> | Tokenflow   | 16384         | 16x16       | 1.37 | 21.41| 0.69 |
> | SDE(MUSE-VL)| 16384         | 16x16       | 2.26 | 20.14| 0.65 |
> | UniTok      | 16384x8       | 16x16x8     | 0.41 | 27.28| 0.77 |
> | QLIP-B      | $ 2^{28}$         | 16x16       | 3.21 | 21.34| 0.59 |
> | **SemHiTok**| 196608        | 16x16       | 1.16 | 21.38| 0.69 |
>
> **Table R1: Comparative experiments on the ImageNet-50K reconstruction task**
>
> > **W2: Experiment of unified MLLM**
>
> **A2:** We appreciate the reviewer’s concern regarding the experiment of unified MLLM.
> We would like to clarify that **Section 2.3 and Supplementary 8.2 of the initial submission describe how the Unified MLLM is deployed with SemHiTok**. In addition, **quantitative and qualitative results are provided in Section 3.3 and Supplementary 8.5 and 8.6**. The **comparisons on multimodal understanding tasks are summarized in Table 2, and generative tasks are presented in Table 3**.
>
> > **W3: More generation results(Gen-Eval, DPG) and some recent method**
>
> **A3:** **In Table R2**, we present the performance on **four generation task benchmarks (DPG, GenEval, GenAI, and MJHQ30K)**, and we have additionally **included several excellent recent methods (Tar, Illume+, Ming-Lite-Uni, UniTok, MMaDA, QLIP-B, and UniFork)**. As shown in the table, **our method achieves competitive results across multiple benchmarks.**
>
> | Method  | DPG$\uparrow$   | GenEval$\uparrow$ | GenAI-Baisc$\uparrow$ | GenAI-Advance$\uparrow$ | MJHQ30K$\downarrow$ |
> | :-----  | :-----: | :-----:| :-----:     | :-----:       | :-----: |
> | Ming-Lite-Uni | /| 0.62    | / | / | / |
> | Transfusion | /   | 0.63   | / | / | / |
> | SynerGen-VL | /   | 0.61   | / | / | 6.10 |
> | VAR     | 71.08   | 0.53   | / | / | / |
> | Show-o  | 67.27   | 0.53   | 0.70        |  0.60         | 15.18   |
> |LlamaGen | 64.84   | 0.32   | / | / | / |
> | EMU3    | 80.60   | 0.66   | / | / | / |
> |QLIP-B   | 78.17   | 0.48   | / | / | / |
> |MUSE-VL  | /       | 0.57   | / | / | / |
> | ILLUME  | /       | 0.61   | 0.75        | 0.60          | 7.76    |
> | ILLUME+ | /       | 0.72   | 0.72        | 0.71          | 6.00    |
> |Tar-1.5B |  82.96 | 0.76 | / | /| /|
> |Tar-1.5B w/Self-Reflect |  84.10 | 0.78 | / | /| /|
> |Tar-7B |  84.19 | 0.84 | / | /| /|
> |Tar-7B w/Self-Reflect |  84.65 | 0.85 | / | /| /|
> | UniFork | /       | 0.46   | /           | /             | 10.60   |
> |TokenFlow| 73.38   | 0.63   | 0.65        | 0.65          | 7.78    |
> |Liquid   | 83.45   | 0.55   | 0.83        | 0.65          | 5.47    |
> |UniTok   | 83.45   | 0.59   | 0.85        | 0.67          | 7.46    |
> |MMaDA    | 69.97   | 0.63   | /        | /     | /  |
> |**SemHiTok** | 83.59 | 0.71 | 0.83        | 0.64          | 5.40    |
>
> **Table R2: Comparison with other methods on generation tasks**

---

### Official Review · Reviewer_HDbp · 2025-10-29

**Soundness:** 2
**Presentation:** 2
**Contribution:** 2
**Rating:** 4
**Confidence:** 4

**Summary:**

The paper proposes SemHiTok, an image tokenizer for unified MLLMs, which provides both semantic features and pixel-level features for multimodal understanding and generation tasks. The tokenizer features a hierarchical architecture, where an image is first quantized into semantic codes and then the pixel-level tokens are selected based on the corresponding pixel sub-codebook. Experiments demonstrate the effectiveness of SemHiTok on both image understanding & generation tasks.

**Strengths:**

* The concept of the hierarchical codebook for high-level semantic features and low-level pixel features is novel, and well-motivated. The method decouples the optimization of the two conflicting objectives for semantics and pixel-level details.
* There are comprehensive experiments which effectively demonstrate SemHiTok's strong performance both at the tokenizer level and in its application on a unified MLLM.

**Weaknesses:**

* While the selection of pixel sub-codebooks is guided by the semantic codes, the training of the pixel-branch is decoupled, so the link between the two hierarchical levels is weak. The authors argue that SemHiTok is superior to naively combining two separately trained tokenizers (in Table 6 Exp 3 & 4), but the conceptual difference feels incremental. Perhaps the authors should provide a more explicit discussion on the specific advantages of this hierarchical design over a simpler, two-stage concatenation approach.

* Though the authors state that SemHiTok avoids codebook overexpansion, the codebook size of SemHiTok is still large (196608), much larger than baseline methods. So the comparison of performances may not be fair, since a much larger codebook inherently allows for higher-fidelity reconstruction.
* There are several grammatical errors, and maybe some mistakes in the formulations (Eq 2, 5).

**Questions:**

About the observation that image patches with the same semantic code tend to have similar pixel feature. Are there any quantitative verification (e.g., specific metrics or analysis) for this observation? This would better solidify the motivation for using the semantic code to index pixel sub-codebooks

---

> ### Author Response · Authors · 2025-11-24
> **Reply to Reviewer HDbp (1/2)**
>
> We thank reviewer HDbp for their valuable feedback. We are glad that you give **"novel, and well-motivated"** on our method, and recognize that our experiments **"effectively demonstrate SemHiTok's strong performance"**. We have addressed your questions as follows.
>
> > **W1: More discussion about the specific advantages of this hierarchical design over a simpler, two-stage concatenation approach**
>
> **A1:** Thank you for your insightful observations and comments on SemHiTok. To address your concerns, we provide the following clarifications:
>
> 1. In SemHiTok, **the semantic codebook and the pixel sub-codebook are decoupled during training but strongly correlated in structure, with pixel-space modeling 𝑃(Texture∣Semantic)**. Through this hierarchical structure, SemHiTok better balances semantic information and pixel-level details.
>
> 2. Simple concatenation leads to feature interference, whereas **SemHiTok’s pixel tokens convey conditional refinement information**. Moreover, concatenating the outputs of two independent tokenizers along the feature dimension **results in a codebook size that scales multiplicatively ($K_{sem}\times K_{pix}$)**, which limits its applicability in unified MLLMs.
>
> > **W2: Ablation of codebook size**
>
> **A2:** Thank you for this insightful observation. We are pleased to offer additional clarification.
> 1. **The actual capacity of the codebook.** The 196k codebook size in SemHiTok refers to the flattened code count used to make processing more convenient for the LLM. In fact, **for VILA-U, which uses a residual quantization codebook, the effective codebook capacity is $16384^4$**, which is far larger than the linear space of SemHiTok (K×m). However, as shown in Tables 1 and 2 of the paper, VILA-U performs worse than SemHiTok in both reconstruction and multimodal understanding.
> 2. To facilitate a more equitable comparison, we **conduct additional ablation experiments using different K×M**, while keeping the total codebook size consistent with the baseline, as shown in Table R1. Since simple codebook structures tend to suffer from reduced utilization when the codebook size becomes too large, **the baseline methods we compare against are all specifically designed to improve codebook utilization.** As shown in **Table R1**, under the same codebook size, **our method remains competitive compared to these expert models for reconstruction tasks**.
>
> | Method         | Codebook Size   | rFID $\downarrow$ | Usage $\uparrow$ |
> | --------       | --------        | ---- | ---   |
> | VQGAN-LC       | 65536           | 2.63 | 100.0 |
> | VQGAN-LC(CLIP) | 65536           | 2.40 | 100.0 |
> | FSQ            | 64000           | 2.80 | 100.0 |
> | LFQ            | 65536           | 2.88 | 100.0 |
> | SimVQ          | 65536           | 2.24 | 100.0 |
> | **SemHiTok**   | 8192x8(65536)   | 1.93 | 98.1  |
> | **SemHiTok**   | 16384x4(65536)  | 1.84 | 99.0  |
> | | | | |
> | SimVQ          | 262144          | 1.99 | 100.0 |
> | Open-MAGVIT2   | 262144          | 1.17 | 100.0 |
> | IBQ            | 262144          | 1.00 | 84.0  |
> | **SemHiTok**   | 8192x32(262144) | 1.21 | 83.7  |
> | **SemHiTok**   | 16384x16(262144)| 1.19 | 79.3  |
>
> **Table R1: Comparison with other methods on ImageNet-50K under different codebook sizes.**
>
> > **W3: Typo**
>
> **A3:** We will make the revisions in the revised version of the paper.

---

> ### Author Response · Authors · 2025-11-24
> **Reply to Reviewer HDbp (2/2)**
>
> > **Q1: Quantitative verification of the same semantic code tends to have similar pixel features**
>
> **A4:**  We sincerely thank you for your insightful and constructive comments.
> Since **there has been no prior work that quantitatively analyzes this phenomenon**, we **attempt to evaluate it using the following approach**:
> 1. Using the tokenizer to extract the code representations along with their corresponding image patches.
> 2. Applying the **DCT(Discrete Cosine Transform)** to extract patch's pixel features.
> 3. For each code, we compute the variance of the DCT features extracted from its associated image patches, followed by **averaging these variances over all codes, denoted as $V_{\text{mean}}$**.
> 4. Computing the **global variance of all image patches, denoted as $V_{\text{global}}$**.
> 5. Computing **Variance Reduction Ratio (VRR)**, denoted as $VRR=1 - V_{\text{mean}}$ / $V_{\text{global}}$. **The larger VRR indicates that the patches corresponding to the given code have more similar pixel features**.
>
> Moreover, we also deploy two baselines:
> * **Random Baseline (Lower Bound)**: Randomly replacing the code corresponding to image patches, the computed result represents the global noise level.
> * **VQGAN Baseline (Reference Ceiling)**: Since VQGAN is a tokenizer designed for pixel-level reconstruction, it can serve as an upper bound.
>
> We **randomly sampled 10K images from the ImageNet-Val (50K) dataset across 5 independent runs**. As shown in Table R2:
> 1. $SemHiTok_{sem}$ is significantly higher than the Random Baseline, demonstrating that **same semantic code tend to have similar pixel feature**;
> 2. VQGAN substantially outperforms $SemHiTok_{sem}$, indicating **the limitation of a semantics-only tokenizer in capturing pixel-level information**;
> 3. $SemHiTok_{pix}$ significantly surpasses both VQGAN and $SemHiTok_{sem}$, showing that **the pixel sub-codebook can effectively model pixel-level information**.
>
> | Random BaseLine | $SemHiTok_{sem}$ | VQGAN | $SemHiTok_{pix}$ |
> | :----: | :--: | :----: | :--: |
> | 0.185±0.005% | 5.245±0.085% | 9.535±0.095% | 13.705±0.075% |
>
> **Table R2: VRR comparison experiments across different tokenizers**

---

### Official Review · Reviewer_JSAk · 2025-11-01

**Soundness:** 3
**Presentation:** 4
**Contribution:** 3
**Rating:** 6
**Confidence:** 3

**Summary:**

This paper proposes SemHiTok, a unified image tokenizer enhanced by semantic information guidance. It innovatively introduces a hierarchical codebook structure, which builds a pixel sub-codebook based on a pre-trained semantic codebook. The semantic part and the pixel part are trained separately to decouple the structure and training strategy. This enables the tokenizer to capture pixel features while retaining its ability to comprehend high-level semantic information. Additionally, SemHiTok is applied to the MLLM structure, and the experimental results demonstrate the performance of this method.

**Strengths:**

- The experiments are comprehensive, comparing with multiple state-of-the-art (SOTA) models and demonstrating the superiority of the SemHiTok method.
- The writing language is accessible, and the diagrams are clear.
- As a Tokenizer, it underwent complete training on the MLLM architecture, validating its effectiveness.

**Weaknesses:**

- The paper only conducted ablation experiments on the joint training vs. phased training of the SemHiTok architecture, but did not compare joint training and phased training across other methods. Thus, the conclusion that "Joint training degrades performance" lacks sufficient support.
- The roles of the sub-codebook and phased training have not been ablated individually in the experiments, making it insufficient to demonstrate the degree of validity of each component on its own.

**Questions:**

- Typo: In the header of Table 5, "Ehance" should be corrected to "Enhance".
- How is the size m of the sub-codebook determined, and does it impact the model's performance?
- Is it necessary to retrain the entire LLM to verify SemHiTok's capabilities? Perhaps one can only replace the Tokenizer in the existing MLLM (Multimodal Large Language Model) structure. Is this new MLLM architecture necessary?

---

> ### Author Response · Authors · 2025-11-24
> **Reply to Reviewer JSAk**
>
> We deeply thank Reviewer JSAk for their valuable opinion and constructive feedback on our work. We are encouraged that you recognize the **"experiments are comprehensive"**, and our method achieves **"comparing with multiple state-of-the-art (SOTA) models "**. We have addressed your questions as follows.
>
> > **W1: The ablation of the staged training process on other methods**
>
> **A1:** We appreciate the reviewer’s suggestion regarding the ablation of the staged training process. To address this, we **further clarify the structures of existing unified tokenizers**.
>
> For example, the **core module Shared-Mapping proposed in TokenFlow** is required to process both discretized semantic features and pixel features simultaneously. **VILA-U adopts an RQ codebook with shared parameters** for the discrete modeling of semantic and pixel features. **These modules do not support phased training, and therefore, we conducted ablation experiments only within our own framework**. In SemHiTok, our proposed SGHC effectively balances semantic and texture information, and **based on this design, decoupled training proves to be more stable**.
>
> > **W2: The ablation of sub-codebook and phased training**
>
> **A2:** We acknowledge the reviewer’s concern regarding the sub-codebook and phased training.
>
> For the sub-codebook, we **present its effectiveness in Table 5 of the initial submitted version**. From the first row of Table 5, we can see that using only the semantic codebook performs well on multimodal understanding tasks, but it is unable to accomplish pixel reconstruction, with an rFID of only 3.17. After adding SGHC, the model’s multimodal understanding ability is almost unaffected, but its pixel reconstruction performance improves significantly, with the rFID reaching 1.42. **To further analyze the impact of sub-codebooks on model performance, we additionally conducted ablation studies on K  and m, as shown in Tables R1 and R2**.
>
> For phased training, we **conducted ablation studies under the SemHiTok framework, as shown in Table 6 in the initial submitted version**. In Table 6, **Exp2** adopts the SemHiTok architecture but uses joint training. Its codebook utilization is only 45.9%, and its performance on both multimodal understanding and reconstruction tasks shows a clear decline compared with SemHiTok(**Exp4**)
>
> | **Method**  | POPE$\uparrow$ | MME-P$\uparrow$  | SEED$\uparrow$ | GQA$\uparrow$ | rFID$\downarrow$ |
> | --------    | :--: | :----: | :--: | :--: | :---:|
> | LlamaGen    | 65.6 | 716.8  | 35.0 | 39.8 | 2.16 |
> | Emu3.5(IBQ) | 73.8 | 904.7  | 40.3 | 46.9 | 0.46 |
> | SDE(MUSE-VL)| 77.3 | 1240.0 | 56.7 | 58.0 | 2.26 |
> | VILA-U      | 81.6 | 1311.6 | 56.9 | 55.3 | 1.80 |
> | **SemHiTok(K=4096,wo pixel branch)** | 79.7 | 1297.7 | 60.4 | 56.6 | / |
> | **SemHiTok** | 82.5| 1355.8 | 62.9 | 60.3 | 1.16 |
>
> **Table R1: Comparative experiments of Emu3.5 and other unified tokenizers on multimodal understanding tasks.**
>
> | K     | m  | rFID$\downarrow$ | Usage$\uparrow$|
> | ---   | -- | ---  | --   |
> | 8192  | 12 | 1.58 | 95.5 |
> | 8192  | 16 | 1.45 | 92.7 |
> | 8192  | 24 | 1.31 | 89.8 |
> | 8192  | 32 | 1.22 | 83.7 |
> | 16384 | 8  | 1.42 | 96.4 |
> | 16384 | 12 | 1.26 | 93.7 |
> | 16384 | 16 | 1.19 | 79.3 |
>
> **Table R2: Ablation experiments of K and m.**
>
> > **Q1: Typo**
>
> **A3:** We will make the revisions in the revised version of the paper
>
> > **Q2: Ablation of sub-codebook size M**
>
> **A4:** Due to space limitations, we **place the ablation study on the sub-codebook size m in Table 7 of the appendix in the initial submitted version of the paper**. Furthermore, we conduct ablation studies with a wider range of $(K, m)$ configurations. **As shown in Tables R2 and R3**, **considering multimodal understanding performance, reconstruction quality, and codebook utilization comprehensively**, we select $K = 16384$ and $m = 12$ as the default settings.
>
> | K     | POPE |MME-P   | SEED | GQA  | Usage |
> | --    | --   | --     | ---- | ---  | ---   |
> | 4096  | 79.7 | 1297.7 | 60.4 | 56.6 | 100   |
> | 8192  | 81.9 | 1343.0 | 61.7 | 59.7 | 99.7  |
> | **16384** | 82.5 | 1355.8 | 62.9 | 60.3 | 99.0 |
> | 32768 | 82.8 | 1364.2 | 62.5 | 60.6 | 93 |
>
> **Table R3: Ablation study on semantic codeboke size K.**
>
> > **Q3: Is directly replacing the existing tokenizer a feasible approach?**
>
> **A5:** Thanks for your comment. We would like to clarify that **deploying unified MLLM based on SemHiTok does not require modifying the MLLM architecture**, only the vocabulary and the language modeling head layer need to be extended. Moreover, **due to differences in feature distributions, codebook sizes, and embedding dimensions across tokenizers, retraining is necessary**.

---

> ### Comment · Reviewer_JSAk · 2025-11-27
>
> Thanks for your comprehensive rebuttal and the detailed supplementary experiments. I will maintain my original score.

---

### Official Review · Reviewer_YG9D · 2025-11-01

**Soundness:** 2
**Presentation:** 3
**Contribution:** 2
**Rating:** 6
**Confidence:** 5

**Summary:**

This paper introduces SemHiTok, a unified image tokenizer designed to effectively capture both high-level semantic features for understanding and low-level pixel details for generation. The authors identify a key challenge in joint training: the inherent conflict between the feature priorities of multimodal understanding and generation tasks. To bridge this gap, they propose a novel Semantic-Guided Hierarchical Codebook (SGHC), which employs a set of sub-codebooks to model the pixel-level space under the guidance of each semantic code. A notable advantage of SemHiTok is its compatibility with existing next-token-prediction-based MLLMs, achieved through a straightforward codebook flattening operation.

**Strengths:**

This paper is well-structured and clearly written, making it a pleasure to read. The core idea of SemHiTok is both novel and compelling, presenting a fresh perspective on the problem. Consequently, I have no major questions regarding the technical content presented. Should I have overlooked any aspect of the work, I welcome the authors to clarify it in their rebuttal.

**Weaknesses:**

Although this paper proposes an alternative method to bridge the gap between low-level visual cues and high-level semantic features, I still have several concerns.

First, the hierarchical structure significantly increases the codebook size. As shown in Table 1, SemHiTok has a total size of K × M = 196,000. It appears impractical to expand this codebook further. However, this already large size limits the value of K. I wonder whether a small K is sufficient to represent the full diversity of semantics in visual content.

Second, the comparisons regarding codebook capacity do not seem entirely fair. As shown in the table, methods like FQGAN and IBQ achieve better reconstruction quality with the same resolution and codebook dimension, yet have a smaller codebook size. This raises a question: given abundant data (far beyond ImageNet-50K), would simply using a large enough standard codebook be sufficient, thereby diminishing the necessity of the proposed SemHiTok? In other words, can SemHiTok maintain its competitiveness under such conditions? For instance, the EMU series (e.g., the recently released EMU-3.5) employs a very large visual codebook and massive pre-training data, which leads to strong performance.

I would appreciate it if the authors could address these concerns.

**Questions:**

* Does the authors mention the specific value of K and m for the SemHiTok's codebook? Have you ever carried out ablation of different value sets of (K, m) and their effectiveness? It could bring more insights if the authors could provide more details about how they decide the scale of codebook.

---

> ### Author Response · Authors · 2025-11-24
> **Reply to Reviewer YG9D (1/2)**
>
> Dear Reviewer YG9D, We appreciate your constructive comments. We are glad that you recognize the **" well-structured and clearly written"**, and our method being able to  **"both novel and compelling"**. We have addressed your questions as follows.
>
> > **W1: Ablation of semantic-codebook size K.**
>
> **A1:** From **Table R1**,
> 1. Compared to LlamaGen, although (K=4096) is much smaller than LlamaGen’s codebook (16,384), the model with (K=4096) still achieves significantly better performance on multimodal understanding tasks, **underscoring the importance of semantic reconstruction training**.
>
> 2. **As the size of the semantic codebook ($K$) increases, the performance on multimodal understanding tasks improves progressively**. However, when using 32K semantic tokenizer compared with the default 16k setting in our experiments, the **performance gain on multimodal tasks is not significant**, and the codebook utilization rate decreases. Overall, **considering both performance and efficiency, we select 16K as the semantic codebook size**.
>
> | K     | POPE |MME-P   | SEED | GQA  | Usage |
> | --    | --   | --     | ---- | ---  | ---   |
> | LlamaGen| 65.6| 716.8 | 35.0 | 39.8 | 97.0  |
> | 4096  | 79.7 | 1297.7 | 60.4 | 56.6 | 100   |
> | 8192  | 81.9 | 1343.0 | 61.7 | 59.7 | 99.7  |
> | **16384** | 82.5 | 1355.8 | 62.9 | 60.3 | 99.0 |
> | 32768 | 82.8 | 1364.2 | 62.5 | 60.6 | 93.0 |
>
> **Table R1: Ablation study on semantic codeboke size $K$.**
>
> > **W2: Does SemHiTok still have an advantage over other VQ methods at a reasonably large codebook size?**
>
> **A2:** We appreciate your concern about the SemHiTok framework. We provide further analysis from the following perspectives:
> 1. The **core of SemHiTok is to better balance semantic and pixel-level information**. In contrast, **FQGAN and IBQ are an improvement over LlamaGen-type methods specifically for pixel reconstruction tasks** and are not suitable for multimodal understanding tasks.
> 2. Although **EMU3.5** uses an IBQ codebook and incorporates semantic distillation, its multimodal **understanding ability remains limited**. Even with a **smaller semantic codebook $K=4096$, performance on understanding tasks is clearly better than EMU3.5**, indicating a significant gap between semantic and pixel information and the need to structurally decouple the two.
> 3. **The actual codebook capacity of FQGAN and VILA-U is much larger than that of SemHiTok**. For VILA-U, which uses 4 layers of residual quantization, its codebook capacity is **$16384^4$**, **far greater than 16384×12**. However, as shown in Tables 1 and 2 in the paper, **VILA-U performs worse than SemHiTok in extracting both semantic and pixel information**. This indicates that simply enlarging the codebook capacity does not directly improve the performance of a unified tokenizer. **The key lies in how effectively the codebook is guided to process semantic information and pixel information**.
>
> | **Method**  | POPE$\uparrow$ | MME-P$\uparrow$  | SEED$\uparrow$ | GQA$\uparrow$ | rFID$\downarrow$ |
> | --------    | :--: | :----: | :--: | :--: | :---:|
> | LlamaGen    | 65.6 | 716.8  | 35.0 | 39.8 | 2.16 |
> | Emu3.5(IBQ) | 73.8 | 904.7  | 40.3 | 46.9 | 0.46 |
> | SDE(MUSE-VL)| 77.3 | 1240.0 | 56.7 | 58.0 | 2.26 |
> | VILA-U      | 81.6 | 1311.6 | 56.9 | 55.3 | 1.80 |
> | **SemHiTok(K=4096,wo pixel branch)** | 79.7 | 1297.7 | 60.4 | 56.6 | / |
> | **SemHiTok** | 82.5| 1355.8 | 62.9 | 60.3 | 1.16 |
>
> **Table R2: Comparative experiments of Emu3.5 and other unified tokenizers on multimodal understanding tasks.**
>
> | Method         | Codebook Size   | rFID$\downarrow$ | Usage$\uparrow$ |
> | --------       | :------:        | :--: | :--:  |
> | VQGAN-LC       | 65536           | 2.63 | 100.0 |
> | VQGAN-LC(CLIP) | 65536           | 2.40 | 100.0 |
> | FSQ            | 64000           | 2.80 | 100.0 |
> | LFQ            | 65536           | 2.88 | 100.0 |
> | SimVQ          | 65536           | 2.24 | 100.0 |
> | **SemHiTok**   | 8192x8(65536)   | 1.93 | 98.1  |
> | **SemHiTok**   | 16384x4(65536)  | 1.84 | 99.0  |
> | | | | |
> | SimVQ          | 262144          | 1.99 | 100.0 |
> | Open-MAGVIT2   | 262144          | 1.17 | 100.0 |
> | IBQ            | 262144          | 1.00 | 84.0  |
> | **SemHiTok**   | 8192x32(262144) | 1.21 | 83.7  |
> | **SemHiTok**   | 16384x16(262144)| 1.19 | 79.3  |
>
> **Table R3: Comparison with other methods on ImageNet-50K under different codebook sizes.**

---

> ### Author Response · Authors · 2025-11-24
> **Reply to Reviewer YG9D (2/2)**
>
> > **Q1: The settings for K and M, along with their corresponding ablation studies**
>
> **A3:** We acknowledge the reviewer’s concern regarding the different value sets of (K, m). In the initially submitted version of our paper, we present the effects of m and semantic codebook dimensionality on both reconstruction and understanding tasks in Tables 7 and 8 in the initial submission. To provide a more detailed explanation, we analyze in **Table R1 the impact of K on understanding tasks**, and further elaborate on this in our response to W1.
>
> We also conduct ablation studies on different sets (K, m). **In Table R4**:
> 1. Under the same K, **enlarging the sub-codebook improves reconstruction quality, but also leads to reduced codebook usage**.
> 2. When the total size (K×m) is kept constant, K=16384 performs slightly better than K=8192.
> 3. Considering understanding performance, reconstruction quality, and codebook usage, our main experiments adopt K=16384, m=12 as the default setting.
>
> | K     | m  | rFID$\downarrow$ | Usage$\uparrow$|
> | ---   | -- | ---  | --   |
> | 8192  | 12 | 1.58 | 95.5 |
> | 8192  | 16 | 1.45 | 92.7 |
> | 8192  | 24 | 1.31 | 89.8 |
> | 8192  | 32 | 1.22 | 83.7 |
> | 16384 | 8  | 1.42 | 96.4 |
> | 16384 | 12 | 1.26 | 93.7 |
> | 16384 | 16 | 1.19 | 79.3 |
>
> **Table R4: Ablation experiments of K and m.**

---

### Author Response · Authors · 2025-11-29
**Summary and thanks to all reviewers and ACs**

We sincerely thank all reviewers for their thoughtful and constructive feedback.
We appreciate the recognition of SemHiTok’s **“novel and compelling”** core idea (YG9D), **“well-motivated”** hierarchical architecture (HDbp), and **“comprehensive”** experiments (JSAk, HDbp). Reviewers highlighted that our method offers a **“fresh perspective”** (YG9D) on bridging the gap between high-level semantics and low-level pixel details, effectively **“decouples the optimization”** of conflicting objectives (HDbp), and achieves **“strong performance”** across both image understanding and generation tasks (HDbp, JSAk). Additionally, reviewers commended the paper for being **“well-structured and clearly written”** (YG9D) with **“clear and easy to follow”** illustrations (bysp). We are grateful for the reviewers’ engagement, which has helped us strengthen our ablation studies and broaden our evaluation benchmarks.

---
#### ***Note: The brickred parts in the paper indicate content added in the revised version.***
---
### **Summary of feedback and rebuttal experiments:**

| Reviewer | Rating | Index | Corcern | Response |
| -------- | ------ | ----- | ------- | -------- |
| YG9D     | 6      | W1    | Ablation of semantic-codebook size $K$ | Additional **experiments** illustrating the impact of K on multimodal understanding task.（**Tab.12** in revision paper）&#x2714; |
|      |     | W2  | Whether SemHiTok outperforms other methods at a reasonably large codebook size | Clarifying the **misunderstanding on codebook capacity definition**; Comparison with other methods under smaller $K$;Additional **experiments** demonstrating SemHiTok’s advantage in **reconstruction** tasks under codebooks of **the same capacity**.(**Tab.14** in revision paper) &#x2714; |
| | | Q1 | Ablation of semantic-codebook size $K$ and sub-codebook size $m$ | Additional **experiments** illustrating the **impact of ($K$,$m$)** on reconstruction task.(**Tab.13** in revision paper)&#x2714; |
| JSAk | 6 | W1 | Ablation of the staged training process on other methods | Further clarify the structures of existing unified tokenizers.&#x2714; |
| | | W2&Q2 | The ablation of sub-codebook $m$ and phased training | The phased training experiments are shown in Table 6 of the initial submitted paper;More Additional **experiments** of the **semantic codebook size ($K$, $m$)**.(**Tab.12 and Tab.13** in revision paper)&#x2714; |
| | | Q1 | Typo | Make the revisions in the revised version of the paper |
| HDbp | 4 | W1 | Specific advantages of this hierarchical design over a simpler, two-stage concatenation approach | Point out that **pixel and semantic information are complementary** under the SemHiTok structure; the design of the sub-codebook **avoids the codebook size explosion** brought by product quantization.&#x2714; |
| | | W2 | Ablation of codebook size | Additional **same codebook size comparison experiments** with other large codebook baseline methods.（**Tab.14** in revision paper）&#x2714; |
| | | W3 | Typo | Make the revisions in the revised version of the pape.&#x2714;|
| | | Q1 | Quantitative verification of the same semantic code tends to have similar pixel features | Attempted quantitative analysis using **Variance Reduction Ratio (VRR)**, and the results are consistent with the observations in the paper.（**Sec.3.4 and Tab.5** in revision paper)&#x2714; |
|bysp | 4 | W1 | More reconstruction performance | Additional comparison experiments on **PSNR, SSIM, and rFID metrics**.(**Tab.8** in revision paper)&#x2714; |
| | | W2 |  Experiment of unified MLLM | Point out that there are **already deployment details and experimental results regarding unified MLLM in the initial submission version of the paper**.(**Sec.2.3 and 3.3, Tab.3 and Tab.4** in revision paper) &#x2714; |
| | | W3 | More generation results(Gen-Eval, DPG) and comparison with recent methods. | Added comparison results with **recent methods on GenEval and DPG**, it still achieves competitive performance.（**Tab.9** in the revision paper） &#x2714; |

* **Reviewer's Response**:  As of the point when ICLR closed the reviewer response channel, **Reviewer JSAk** responded to our rebuttal and made it clear that he will **maintain my original score(Rating: 6)**. Other reviewers did not respond.

---
***It is very unfortunate that we encountered an OpenReview bug, which prevented us from having a complete discussion with the reviewers. Our method is now stronger, clearer, and better supported thanks to the feedback. Thank you once again for your time and thoughtful engagement. We look forward to your assessment and will incorporate all final recommendations into the camera-ready version.***

---

### Meta-Review · Area_Chair_meDv · 2026-01-07

**Summary:**

This paper presents a unified image tokenizer with a hierarchical cocebook structure for enhanced semantic information. Reviewers praised the idea and writing, where comprehensive experiments are presented for a fresh perspective on bridging the gap between high-level semantics and low-level pixel details.

**Reviewer Concerns:**

HDbp: W1-3 are addressed, Q1 is addressed. Rating increase anticipated.

bysp: W1 is addressed, W2 is clarified but may remain, W3 is responded with more results.

JSAk: W1 is clarified but may remain. W2 is addressed. Q1-Q3 are addressed.

YG9D: W1 is addressed, W2 is responded but may not be fully addressed, Q1 is addressed.

**Reviewer Scores:**

JSAk: 6 -> 6
YG9D: 6 -> 6
HDbp: 4 -> 6
bysp: 4 -> 4

Average: 5 -> 5.5

---

### Decision · Program_Chairs · 2026-01-26

Accept (Poster)